# FGF2-FGFR1 signaling regulates release of Leukemia-Protective exosomes from bone marrow stromal cells

Nathalie Javidi-Sharifi[1†], Jacqueline Martinez[1†], Isabel English[1], Sunil K Joshi[1], Renata Scopim-Ribeiro[1], Shelton K Viola[1‡], David K Edwards V[1], Anupriya Agarwal[1,2], Claudia Lopez[1,3], Danielle Jorgens[1,3], Jeffrey W Tyner[1,4], Brian J Druker[1,2,5], Elie Traer[1,2]*

[1]Knight Cancer Institute, Oregon Health & Science University, Portland, United States; [2]Division of Hematology and Medical Oncology, Oregon Health & Science University, Portland, United States; [3]Center for Spatial Systems Biomedicine, Oregon Health & Science University, Portland, United States; [4]Department of Cell, Developmental & Cancer Biology, Oregon Health & Science University, Portland, United States; [5]Howard Hughes Medical Institute, Chevy Chase, United States

**\*For correspondence:**
traere@ohsu.edu

[†]These authors contributed equally to this work

**Present address:** [‡]Department of Pediatrics, Division of Pediatric Hematology-Oncology, Naval Medical Center Portsmouth, Portsmouth, United States

**Abstract** Protective signaling from the leukemia microenvironment leads to leukemia cell persistence, development of resistance, and disease relapse. Here, we demonstrate that fibroblast growth factor 2 (FGF2) from bone marrow stromal cells is secreted in exosomes, which are subsequently endocytosed by leukemia cells, and protect leukemia cells from tyrosine kinase inhibitors (TKIs). Expression of FGF2 and its receptor, FGFR1, are both increased in a subset of stromal cell lines and primary AML stroma; and increased FGF2/FGFR1 signaling is associated with increased exosome secretion. FGFR inhibition (or gene silencing) interrupts stromal autocrine growth and significantly decreases secretion of FGF2-containing exosomes, resulting in less stromal protection of leukemia cells. Likewise, *Fgf2* -/- mice transplanted with retroviral BCR-ABL leukemia survive significantly longer than their +/+ counterparts when treated with TKI. Thus, inhibition of FGFR can modulate stromal function, reduce exosome secretion, and may be a therapeutic option to overcome resistance to TKIs.

**Editorial note:** This article has been through an editorial process in which the authors decide how to respond to the issues raised during peer review. The Reviewing Editor's assessment is that all the issues have been addressed (see decision letter).
DOI: https://doi.org/10.7554/eLife.40033.001

## Introduction

TKIs have revolutionized the treatment of CML and have shown promise in AML, however development of resistance remains a problem. In CML, resistance develops in a minority of patients, and is most often caused by resistance mutations. However, some patients still develop resistance in the absence of known resistance mutations. In contrast, development of resistance in AML is the norm. Inhibitors of mutated FLT3, which is present in about 30% of AML patients, are initially quite efficacious (*Smith et al., 2012*). However, resistance to FLT3 kinase inhibitors in AML typically develops within a few months. In some cases, resistance is cell-intrinsic and due to secondary mutations in the activating loop of FLT3 that prevent drug binding (*Weisberg et al., 2009*), however, resistance still develops in the absence of these mutations. Within the marrow microenvironment, leukemia cell survival can be mediated by extrinsic ligands that activate alternative survival pathways (*Smith et al.,*

**eLife digest** Leukemias are cancers of white blood cells. The cells grow and divide rapidly, often because of mutations in proteins called kinases. Since the kinase mutations do not occur in healthy cells, they provide a good target for anti-leukemia drugs. Several such kinase inhibitors are effective at treating leukemia patients. However, most leukemia cells develop ways to resist the effects of the kinase inhibitors over time, leading to relapses of the disease.

One way that leukemia cells resist kinase inhibitors is by taking advantage of signals coming from supportive cells, known as stromal cells, in the bone marrow. When patients are treated with kinase inhibitors, the bone marrow stromal cells produce more of a signaling protein called FGF2. The leukemia cells then use FGF2 to survive the effects of the kinase inhibitors.

It was not clear how the FGF2 signal reaches the leukemia cells from the bone marrow stromal cells. Now, using biochemical techniques, Javidi-Sharifi, Martinez et al. show that bone marrow stromal cells package FGF2 into small compartments called exosomes. The stromal cells release the exosomes into the bone marrow, and the leukemia cells then engulf and internalize the exosomes. Leukemia cells that had taken up FGF2 in this way were better able to survive kinase inhibitor treatment than leukemia cells that had not.

Javidi-Sharifi, Martinez et al. also observed that FGF2 also affects the bone marrow stromal cells themselves, causing them to grow faster, produce more FGF2 and release more exosomes. Blocking the effects of FGF2 on the stromal cells slowed their growth and caused fewer exosomes to be released. In addition, mice whose bone marrow stromal cells could not produce FGF2 survived leukemia for longer than mice whose stromal cells provided protective FGF2 in exosomes to leukemia cells. This suggests that taking advantage of drugs that prevent bone marrow stromal cells from releasing FGF2 in exosomes might improve treatments for leukemia. Further research will be needed to confirm whether this strategy would be effective in humans.

DOI: https://doi.org/10.7554/eLife.40033.002

*2017*; *Ghiaur and Levis, 2017*; *Wilson et al., 2012*) and over time can lead to development of intrinsic resistance mutations (*Wilson et al., 2012*; *Traer et al., 2012*).

Bone marrow stromal cells provide a supportive structure and secrete cytokines that contribute to the normal hematopoietic stem cell niche, but can also protect leukemic cells from therapy (*Colmone et al., 2008*; *Ayala et al., 2009*). Initial studies into the mechanisms of resistance utilized normal marrow stroma (*Manshouri et al., 2011*), but the stroma can be altered by leukemia, in a manner similar to development of cancer associated fibroblasts in solid tumors (*Paggetti et al., 2015*; *Huang et al., 2015*; *Huan et al., 2015*). We found that fibroblast growth factor 2 (FGF2) expression is increased in marrow stromal cells during tyrosine kinase inhibitor (TKI) therapy and protects leukemia cells (*Ware et al., 2013*; *Traer et al., 2014*; *Javidi-Sharifi et al., 2015*; *Traer et al., 2016*). FGF2 has also been shown to be essential for stress hematopoiesis after chemotherapy (*Itkin et al., 2012*; *Zhao et al., 2012*), suggesting that leukemia cells can hijack a normal marrow stress response for their own survival.

Despite its important roles in physiology and pathology, several aspects of FGF2 biology remain poorly understood. FGF2 does not have a signal peptide and is not secreted through the canonical secretory pathway. Alternative mechanisms for secretion have been proposed, but how FGF2 is conveyed between two cells remains unclear (*Steringer et al., 2015*; *Zacherl et al., 2015*). Additionally, while recombinant FGF2 directly stimulates myeloid colony formation (*Berardi et al., 1995*), there are also reports suggesting that FGF2 can indirectly regulate hematopoiesis by stimulating stromal cells to produce cytokines (*Avraham et al., 1994*).

We discovered that FGF2 is largely secreted in extracellular vesicles (ECVs) and exosomes from bone marrow stromal cells. ECVs are able to protect leukemia cells from the effects of TKI therapy. Furthermore, autocrine FGF2-FGFR1 activation in marrow stromal cells increases the secretion of FGF2-laden exosomes, indicating that exosome secretion is regulated in part by FGF2-FGFR1 signaling. Inhibition of FGFR1 can reverse this protective stroma-leukemia interaction and restore leukemia cell TKI sensitivity in the marrow niche.

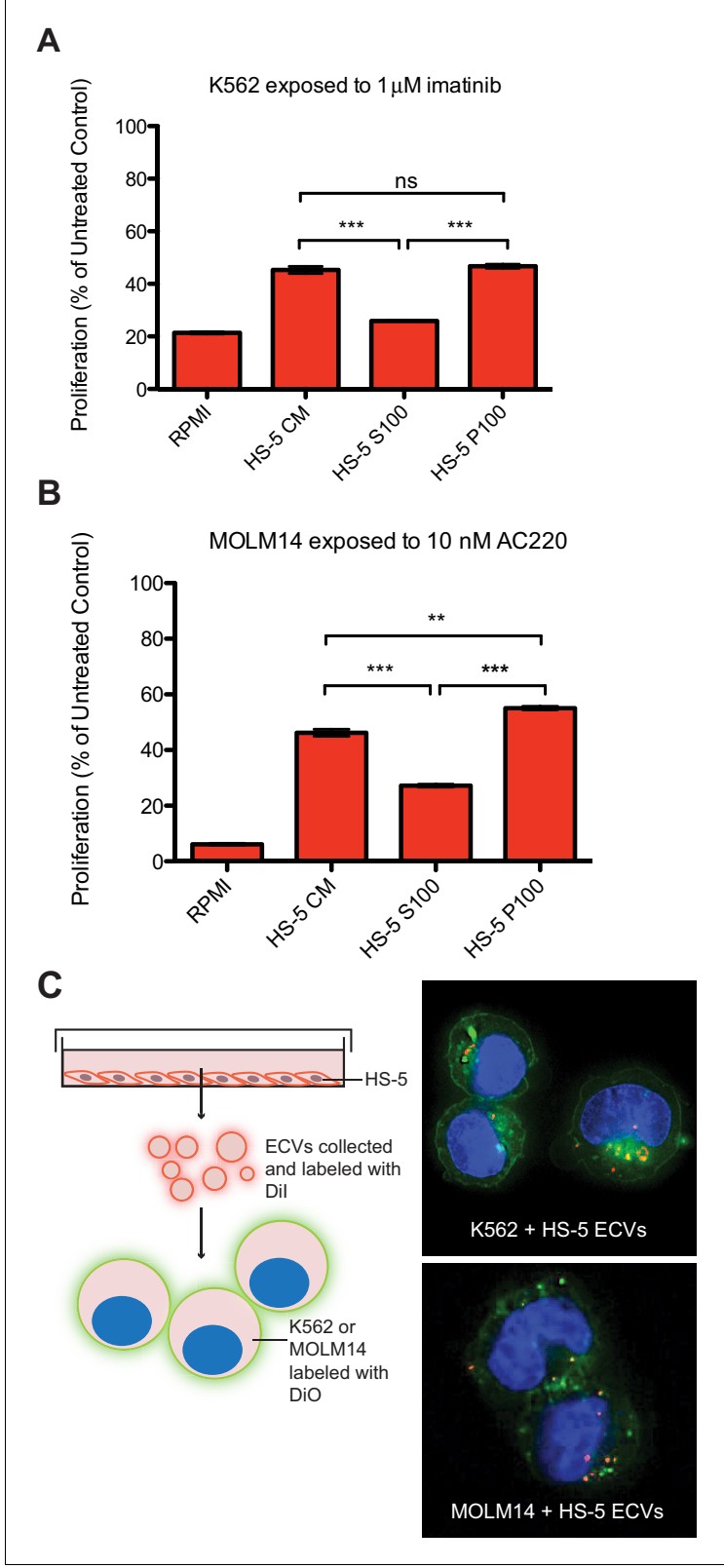

**Figure 1.** Extracellular vesicles (ECVs) secreted by HS-5 cells are internalized by MOLM14 and K562 cells and protect from treatment with AC220 or imatinib, respectively. HS-5 conditioned media (CM) was collected and separated by ultracentrifugation at 100,000 g into a supernatant (S100) and pellet (P100) fraction containing ECVs. These fractions were incubated with (A) K562 cells ± 1 μM imatinib, or (B) MOLM14 cells ± 10 nM AC220, and

*Figure 1 continued*

viability measured by MTS assay after 48 hr. Values were normalized to respective untreated condition. All wells were plated in triplicate and error bars indicate standard deviation. RPMI is the media control. p values are indicated by *<0.05, **<0.005, and ***=0.0007. (C), MOLM14 and K562 cells were stained with DiO (green) tracer, washed, and immobilized on Poly-D-Lysine coated chamber slides. HS-5 P100 fraction was stained with DiI (red) tracer and added to the cells for a 24 hr incubation. Slides were stained with DAPI (blue) and imaged by confocal fluorescent microscopy. A movie of the z-stack images is included in Supplemental data.

DOI: https://doi.org/10.7554/eLife.40033.003

The following video and figure supplement are available for figure 1:

**Figure supplement 1.** Comparison of protection from recombinant FGF2, HS-5 ECVs, and CM after ECV depletion (-ECV) in both K562 and MOLM14 cells.

DOI: https://doi.org/10.7554/eLife.40033.004

**Figure 1—video 1.** HS-5 ECVs were stained with DiI (red) and K562 cells were stained with DiO (green) and incubated for 24 hr at 37°C as described in Materials and methods.

DOI: https://doi.org/10.7554/eLife.40033.005

**Figure 1—video 2.** HS-5 ECVs were stained with DiI (red) and MOLM14 cells were stained with DiO (green) and incubated for 24 hr at 37°C as described in Materials and methods.

DOI: https://doi.org/10.7554/eLife.40033.006

## Results

### Stromal cell ECVs protect leukemia cells from TKI therapy

The human stromal cell line HS-5 expresses abundant FGF2, in addition to other soluble cytokines such as IL-5, IL-8 and HGF (*Roecklein and Torok-Storb, 1995*), and conditioned media (CM) from HS-5 is highly protective of leukemia cell lines. HS-5 CM was ultracentrifuged at 100,000 g to separate soluble proteins (supernatant, S100) from ECVs and larger macromolecules (pellet, P100). We compared the protective effect of unfractionated CM, S100, and P100 fractions on the viability of two leukemia cell lines: MOLM14 (FLT3 ITD+ AML) and K562 (CML), in the presence of their respective TKIs, quizartinib (AC220, a highly selective and potent inhibitor [*Zarrinkar et al., 2009*]) and imatinib (*Figure 1A and B*). The protective capacity of the S100 fraction was less than unfractionated CM, and protection was enriched in the concentrated P100 ECV fraction (*Figure 1*), indicating that a substantial protective component of HS-5 CM is mediated by ECVs. A more extensive profiling of protection is also shown in *Figure 1—figure supplement 1*.

To determine if ECVs produced by HS-5 cells are internalized by K562 and MOLM14 leukemia cells, K562 and MOLM14 cells were stained with a green lipophilic tracer (DiO) and incubated with HS-5 ECVs stained with a red lipophilic tracer (DiI). Analysis by confocal microscopy showed that ECVs are indeed internalized by leukemia cells, although the exact mechanism of internalization is still under investigation (*Figure 1C* and *Figure 1—video 1* and *2*).

### FGF2 is contained in stromal cell ECVs and exosomes

FGF2 is highly expressed in the HS-5 stromal cell line but the related HS-27 expresses little FGF2 (*Figure 2A*; *Traer et al., 2016*). We analyzed FGF2 in S100 and P100 fractions of both HS-5 and HS-27 by immunoblot (*Figure 2B*). Little FGF2 was detected in the soluble protein fraction (S100), but FGF2 was enriched in ECVs (P100). Washing the ultracentrifuge tube with detergent liberated even more FGF2 (detergent wash P100), due to ECVs adhering to the ultracentrifuge tube. To compare FGF2 to other soluble cytokines, HS-5 CM was ultracentrifuged into S100 and ECVs, cytokines quantified by Luminex multiplex assay, and normalized to unfractionated CM (*Figure 2C*). Pelleted ECVs were resuspended in 10% of the original CM volume, and the P100 bars in *Figure 2B* thus represent a 10-fold enrichment, although as shown in *Figure 2B* not all ECVs can be liberated from the ultracentrifuge tube. FGF2 was uniquely enriched in ECVs, whereas soluble cytokines such as stem cell factor, interleukin (IL)−6, IL-8, etc. were found primarily in the S100 fraction.

HS-5 ECVs were further separated into microvesicles, exosomes, and insoluble extracellular matrix proteins (ECM) using a sucrose step-gradient to separate by density. FGF2 and cell compartment-specific molecular markers were probed by immunoblot (*Figure 2D*). FGF2 was most highly enriched in the 15–30% sucrose interface, which also contained the exosome-specific marker CD9.

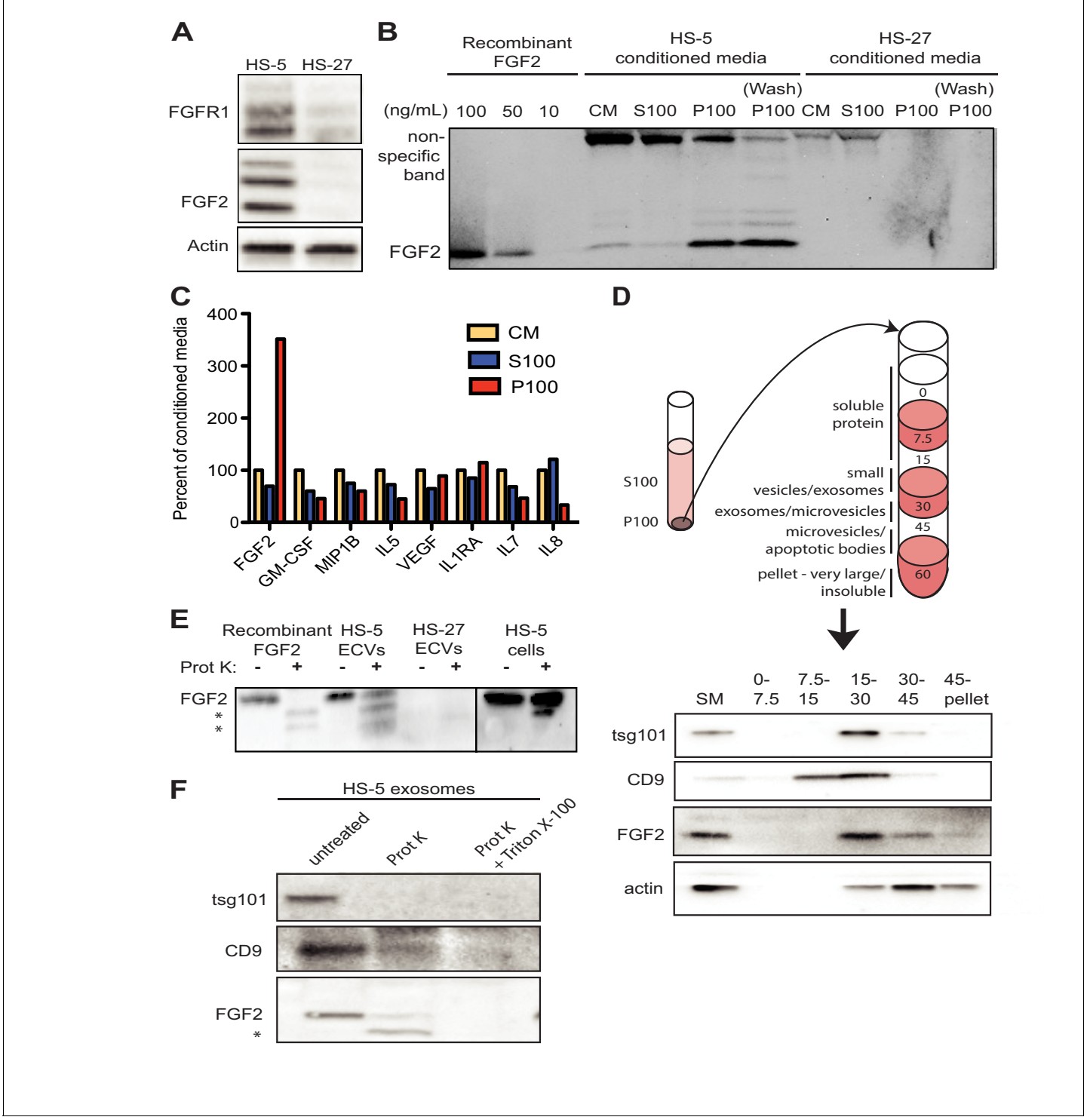

**Figure 2.** FGF2 is enriched in exosomes from HS-5 bone marrow stromal cells. (A) Immunoblot of FGFR1, FGF2 and actin in HS-5 and HS-27 whole cell lysates. (B) HS-5 and HS-27 CM were ultracentrifuged at 100,000 g for 2 hr at four degrees C. CM, soluble protein (S100), and ECV (P100) fractions were collected and analyzed by immunoblot, using 10, 50, and 100 ng/ml recombinant FGF2 for comparison. The ultracentrifuge tube was also washed with detergent to remove adherent ECVs and material (detergent wash P100). (C) HS-5 CM, S100 and P100 fractions (concentrated ~10 fold compared to HS-5 CM) were solubilized in 0.1% NP-40 and analyzed by cytokine multiplex ELISA (Luminex). The S100 and P100 fractions were normalized to CM. (D) The HS-5 P100 fraction (starting material, or SM) was further fractionated on a sucrose step-gradient. Sucrose layer interfaces (0–7.5%, 7.5–15%, 15–30%, 30–45%, and 45%-pellet) were collected, lysed and analyzed by immunoblot with antibodies against the exosomal marker CD9, FGF2, and cytoplasmic marker actin. (E) HS-5 and HS-27 ECVs (P100), recombinant FGF2, and HS-5 cells were exposed to proteinase K and analyzed by immunoblot. (F) HS-5

*Figure 2 continued on next page*

*Figure 2 continued*

exosomes were isolated by sucrose step-gradient (see panel D) and then exposed to proteinase K with or without detergent (0.1% Triton X-100, used to dissolve the lipid membrane). Samples were subjected to immunoblot analysis using antibodies against tsg101, CD9 and FGF2. The * indicates degraded FGF2 after partial proteinase K digestion.

DOI: https://doi.org/10.7554/eLife.40033.007

To determine if FGF2 was bound to the outside of ECVS, or contained within ECVs, proteinase K was used to digest proteins not enclosed by lipid membrane. Recombinant FGF2, HS-5 or HS-27 ECVs, and intact HS-5 cells were incubated with proteinase K and probed for FGF2 by immunoblot (*Figure 2E*). Recombinant FGF2 was completely degraded by proteinase K (* indicates degraded fragments) but intact FGF2 was detected in both HS-5 ECVs and control HS-5 cells. We repeated this experiment using purified HS-5 exosomes and again observed that a fraction of FGF2 was protected from digestion (*Figure 2F*). Addition of 0.1% Triton X-100 disrupted the lipid membrane and resulted in complete digestion of all protein. We found a similar digestion pattern with the exosomal transmembrane proteins CD9 and tsg101. We conclude that FGF2 is contained within ECVs and exosomes, however we cannot exclude that FGF2 may also be on the surface since partial FGF2 degradation was noted in intact HS-5 cells, ECVs and purified exosomes (*Figure 2E–F*).

## HS-5 stromal cells overproduce ECVs

Since HS-5 CM is more protective than HS-27 CM (*Manshouri et al., 2011*; *Traer et al., 2016*; *Weisberg et al., 2008*), we suspected that ECVs may be more numerous in HS-5 CM. We chose several orthogonal methods to quantify vesicles in CM. First, we used nanoparticle tracking analysis to quantify and compare HS-5 and HS-27 ECVs (*Figure 3A*). In parallel, we employed the Virocyt Virus Counter, a flow cytometry-based technique developed to detect viruses, which also works well to quantify ECVs (*Figure 3B*). As a gold standard, negative stain transmission electron microscopy of purified HS-5 and HS-27 exosomes was also used to image and quantify exosomes by counting (30–100 nm diameter with cup-shape appearance characteristic for exosomes, *Figure 3C*). Finally, we used sucrose step-gradient fractionation of HS-5 and HS-27 ECVs to compare cell compartment and exosome-specific markers by immunoblot (*Figure 3D*). Exosomes layer primarily at the 15–30% sucrose interface as indicated by exosomal markers CD9 and tsg-101, and are increased in HS-5 cells compared to HS-27. Interestingly the receptor for FGF2, FGFR1, was also found to localize preferentially with HS-5 exosomes. With all methods, we consistently observed greater than two-fold excess of vesicles produced by HS-5 compared to HS-27 cells (see *Figure 3—figure supplement 1* for additional data). Markers of nucleus (lamin A/C), endoplasmic reticulum (calreticulin) and mitochondria (Bcl-XL) were located in the 45–60% interface containing larger microvesicles and apoptotic bodies.

## FGF2-FGFR1 signaling promotes stromal growth and paracrine protection of leukemia

FGF2 is an autocrine signaling protein for stroma, but recombinant FGF2 also mediates paracrine protection of leukemia cells (*Traer et al., 2016*; *Traer et al., 2014*). Thus there are two potential mechanisms by which FGFR inhibition can attenuate protection of leukemia cells in the marrow microenvironment: (1) FGFR inhibitors block FGF2-mediated paracrine protection at the leukemia cells; and/or (2) FGFR inhibitors interrupt stromal FGF2-FGFR1 autocrine signaling to reduce secretion of protective FGF2-containing exosomes. To compare the relative effect of FGFR inhibition on autocrine and paracrine signaling, HS-5 cells were pre-treated with the FGFR inhibitor PD173074 (*Mohammadi et al., 1998*; *Trudel et al., 2004*) for one week prior to collection of CM. CM collected from HS-5 cells pre-treated with PD173074 was significantly less protective than CM from an equal number of untreated HS-5 cells (*Figure 4A*), providing evidence that interruption of FGF2-FGFR1 signaling affects subsequent protection of leukemia cells. In contrast, addition of PD173074 to untreated HS-5 CM only modestly attenuated protection of MOLM14 cells. We found similar results with K562 cells exposed to imatinib (*Figure 4—figure supplement 1*). Purified ECVs from HS-5 CM, which are enriched in FGF2, were more sensitive to FGFR inhibition (*Figure 1—figure supplement 1*), however pre-treatment of HS-5 cells with PD173074 still had the greatest absolute reduction in protection. These results indicate that FGFR inhibitors overcome protection of leukemia cells

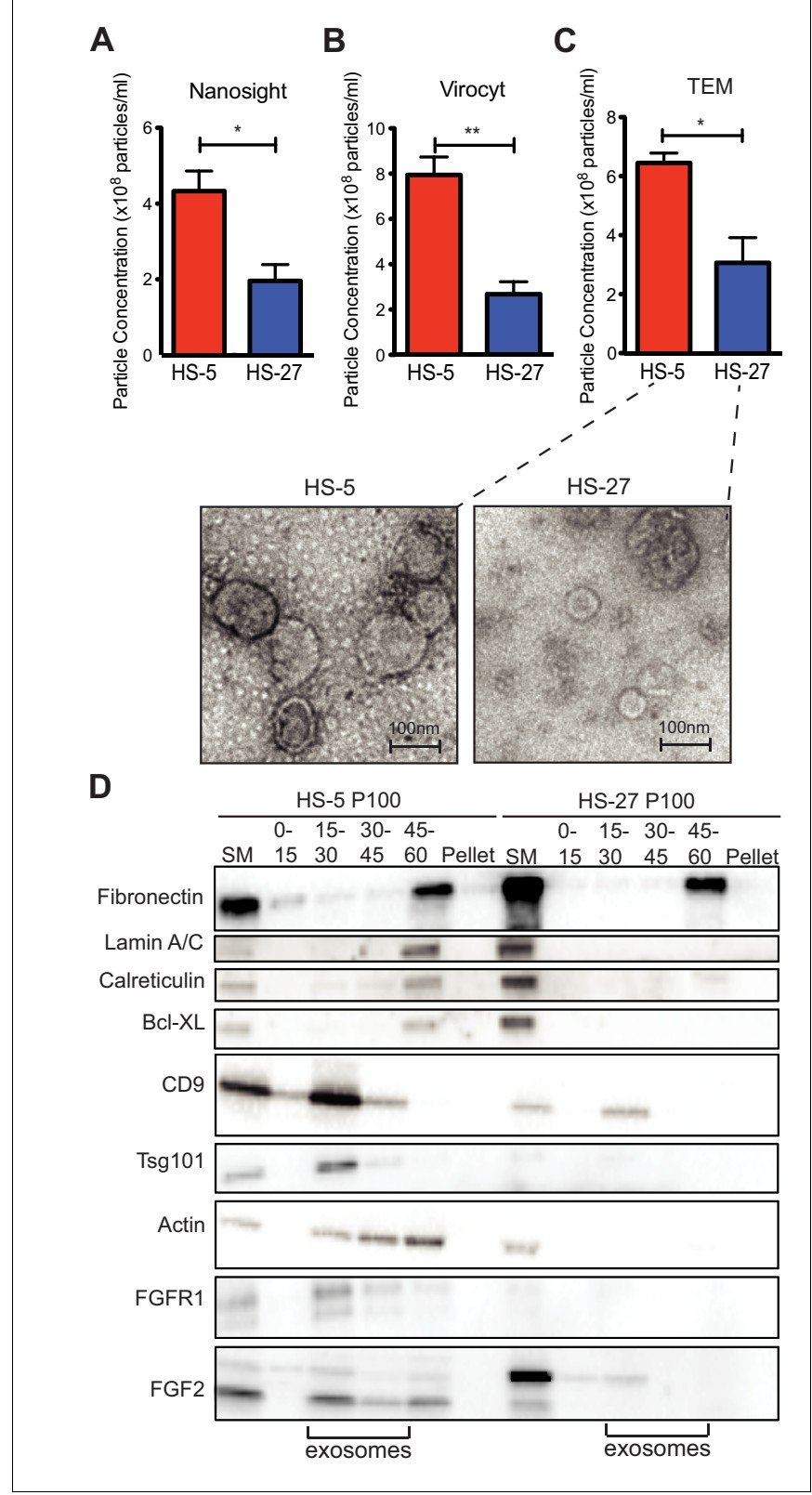

**Figure 3.** HS-5 cells secrete more exosomes than HS-27 cells. Equal numbers of HS-5 and HS-27 cells were plated in RPMI with exosome-depleted FBS for 24 hr. The ECVs were pelleted by ultracentrifugation at 100,000 g for 2 hr at four degrees C and resuspended in PBS. ECVs were quantified by (A) Nanosight, a nanovesicle tracking analysis, (B) Virocyt Virus Counter, a proprietary flow cytometry using fluorescent dyes that stain both nucleic acid

*Figure 3 continued on next page*

*Figure 3 continued*

and protein, or (C) transmission electron microscopy. (D) HS-5 and HS-27 exosomes were collected by sucrose-step gradient and analyzed by transmission electron microscopy. Vesicles were quantified by counting in three 2 × 2 μm areas per sample. All experiments were done in triplicate, error bars represent standard deviation, p values are indicated by *<0.05, **<0.005. HS-5 and HS-27 ECVs (P100) were obtained by ultracentrifugation (starting material, or SM), and the exosome fraction was further purified by a sucrose step-gradient. Sucrose layer interfaces (0–7.5%, 7.5–15%, 15–30%, 30–45%, and 45%-pellet) were collected, lysed and analyzed by immunoblot. Blots were probed with antibodies against exosomal markers CD9 and tsg101; cell compartment markers: fibronectin, lamin A/C, BCL-XL; as well as FGFR1 and FGF2. The lanes with highest enrichment for CD9 and tsg-101, indicating exosomes, are marked below.

DOI: https://doi.org/10.7554/eLife.40033.008

The following figure supplement is available for figure 3:

**Figure supplement 1.** Methods for exosome quantification and further evaluation of microvesicle populations.

DOI: https://doi.org/10.7554/eLife.40033.009

primarily by directly altering secretion of FGF2-expressing stromal cells, making them significantly less protective.

To further evaluate the effects of FGFR inhibition in stromal cells, HS-5 cells were evaluated for viability, morphology, and growth using HS-27 cells as comparison (low FGF2). HS-5 or HS-27 cells had little reduction in cell viability after 72 hr treatment with PD173074 (*Figure 4B*), however HS-5 growth slowed dramatically over 15 days (*Figure 4C*). HS-5 cells exposed to PD173074 changed morphology and became less refractile, larger, and more adherent (*Figure 4D*). Cell size was quantified using CellProfiler software and PD173074 significantly increased HS-5 cell size (*Figure 4E*).

To evaluate FGF2 and FGFR1 expression in primary leukemia stroma, bone marrow aspirates from a series of leukemia patients were cultured ex vivo and FGF2 and FGFR1-4 expression quantified by RT-PCR (*Figure 4F*). FGFR1 and FGF2 transcripts were the most highly expressed in primary stroma, and there was a strong positive correlation between FGFR1 and FGF2 expression (*Figure 4G*, $r^2$ = 0.5683 and p<0.0001 on nonparametric correlation). This indicates that FGF2 and FGFR1 expression are coordinately regulated in primary marrow stromal cells consistent with activation of an FGF2-FGFR1 autocrine loop. There were nine stromal cultures from AML patients with FLT3 ITD (indicated with red dots), but most of them were newly diagnosed, and based upon our previous data we would not expect increased expression of FGF2[16]. Similar to our observations in cell lines described above, we also detected FGFR1 and FGF2 in ECVs derived from primary marrow stromal cultures (*Figure 4—figure supplement 2*). However, primary marrow stromal cells grow slowly and produce smaller amounts of ECVs, so we were unable to evaluate the effect of FGFR inhibitors on cell morphology, growth, and ECV production with primary marrow stromal cells. Additional characterization of primary stromal cultures is contained in *Figure 4—figure supplement 3*.

## FGFR inhibition decreases stromal cell production of exosomes

Since FGFR inhibition attenuates HS-5 growth and morphology, we hypothesized that it might also reduce secretion of ECVs. HS-5 cells exposed to graded concentrations of PD173074 and BGJ-398 had a dose-dependent decrease in ECVs measured by Virocyt Virus Counter (*Figure 5A,B*). Notably, there was a significant decrease in vesicle number as early as 6 hr after drug exposure (*Figure 5—figure supplement 1*), suggesting that FGFR inhibition directly affects vesicle production or release. ECVs were also collected from HS-5 and HS-27 cells exposed to PD173074 and analyzed by immunoblot. PD173074 reduced the exosome markers tsg101 and CD9 (and FGF2) but had no effect on ECV production from HS-27 cells (*Figure 5C*, similar results with BGJ398 shown in *Figure 5—figure supplement 1*). Scanning electron microscopy of HS-5 cells revealed abundant budding membrane, whereas the surface of PD173074-exposed cells was smoother, implicating a change in membrane dynamics (*Figure 5—figure supplement 2*). To evaluate exosome secretion specifically, sucrose step-gradient fractionation was performed on ECVs from untreated and PD173074 treated HS-5 cells. PD173074 reduced exosomal markers CD9, tsg101, and FGF2 in the expected 15–30% interface fraction (*Figure 5D*).

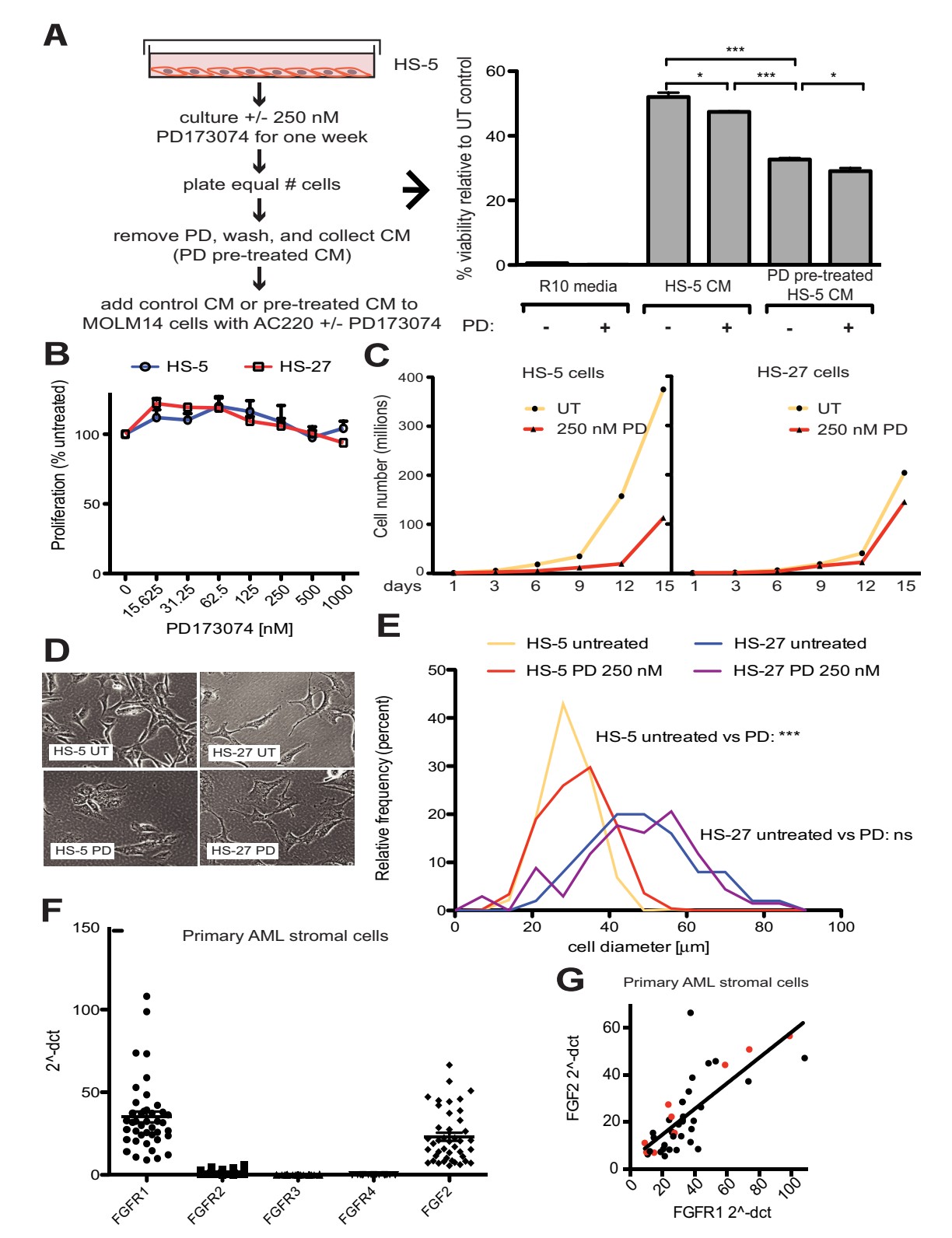

**Figure 4.** FGF2 is an autocrine growth factor in bone marrow stromal cells, and FGFR inhibition attenuates growth. (**A**) HS-5 cells were cultured in media ± 250 nM PD173074 for one week and then equal numbers of cells were replated for comparison. After adhesion, the cells cultured in PD173074 were washed and fresh media added to collect CM. MOLM14 cells were resuspended in media, untreated HS-5 CM, and PD pre-treated HS-5 CM and treated with ± 10 nM AC220 and ± 250 nM PD173074. Viability was measured by MTS assay after 72 hr and values were normalized to the relevant UT

*Figure 4 continued on next page*

Figure 4 continued

control. Error bars represent standard deviation, p values are indicated by *<0.05, **<0.005, and ***=0.0007. (B) HS-5 and HS-27 cells were plated in triplicate on 96 well plates in a gradient of FGFR inhibitor PD173074. Proliferation was measured using MTS reagent after 72 hr. Error bars indicate standard deviation. (C) HS-5 and HS-27 cells were incubated media ± 250 nM PD173074 (PD). The number of viable cells was measured with Guava ViaCount every 3 days over a 15 day period. Fresh media and PD173074 was added every 3 days. (D) HS-5 and HS-27 cells were incubated in media ± 1 µM PD173074 for 1 week. Brightfield microscopy images were obtained using a 10X objective. (E) HS-5 cells were incubated in 4-well glass chamber slides in media ± 250 nM PD173074 (PD). Cells were stained with lipophilic tracer DiI for 24 hr, fixed, then nuclei stained with DAPI. Immunofluorescent images were analyzed with CellProfiler software to determine cell size (µm [*Weisberg et al., 2009*]) and number of cells for each size range was binned and graphically displayed. PD173074 had no effect on HS-27 growth, morphology or size, consistent with an on-target FGFR effect. (F) Ex vivo cultured primary bone marrow stromal cells from a series of leukemia patients (n = 42) were lysed for RNA extraction and cDNA synthesis. Taqman qPCR analysis was performed using FGFR1, FGFR2, FGFR3, FGFR4, and FGF2 Taqman primer assays and expression plotted (n = 42 for each except FGFR4 which is n = 41 due to failed PCR for one sample). (G) FGFR1 and FGF2 qPCR values (2-ΔCT) were plotted against each other. There were 9 AML patients with FLT3 ITD (most newly diagnosed) and these patients are indicated with red dots. Linear regression produced a line fit with $r^2$ = 0.5683 and slope significantly non-zero with p<0.0001.

DOI: https://doi.org/10.7554/eLife.40033.010

The following figure supplements are available for figure 4:

**Figure supplement 1.** Pre-treatment of HS-5 stromal cells with FGFR inhibitor reduces protective properties of HS-5 CM when added to K562 cells exposed to imatinib.
DOI: https://doi.org/10.7554/eLife.40033.011
**Figure supplement 2.** Cultures of primary human and mouse bone marrow stroma produce microvesicles containing FGF2.
DOI: https://doi.org/10.7554/eLife.40033.012
**Figure supplement 3.** Cultured primary human bone marrow stroma exhibits trilineage differentiation.
DOI: https://doi.org/10.7554/eLife.40033.013

## Genetic knock-down of FGFR1 or FGF2 attenuates exosome production

To confirm that decreased exosome secretion is specific for FGFR1 inhibition, HS-5 cells were stably transfected with either a GFP-expressing lentivirus control vector (GIPZ), or doxycycline-induced shRNA targeting FGFR1. FGFR1 silencing led to a significant reduction in ECVs (*Figure 6C*). Similar results were obtained with siRNA targeting FGFR1 (*Figure 6—figure supplement 1*). siRNA and shRNA constructs targeting FGF2 did not achieve reliable silencing of FGF2. HS-5 CRISPR/Cas9 knockout of FGFR1 and FGF2 in HS-5 cells were generated, however genetic silencing prevented continued growth. Multiple attempts to make stable deleted cell lines were unsuccessful, likely due to the importance of FGF2-FGFR1 signaling for HS-5 self-renewal and growth (*Bianchi et al., 2003*; *Coutu et al., 2011*; *Zhou et al., 1998*). That being said, ECVs collected shortly after CRISPR/CAS9 treatment, which resulted in partial silencing of FGF2 or FGFR1, both demonstrated decreased ECVs by immunoblot and reduced protection of MOLM14 cells (*Figure 6—figure supplement 2*). To test the role of FGF2 in ECV production in primary cells, equal numbers of murine stromal cells from *Fgf2* +/+ and -/- mice (*Fgf2tm1Doe* [*Zhou et al., 1998*]) were treated with PD173074 and ECVs quantified by Virocyt (*Figure 6D*). *Fgf2* +/+ stromal cells secreted significantly more ECVs than -/-, and PD173074 only reduced ECV secretion in +/+ stroma. ECVs from *Fgf2* +/+ and -/- mice were also analyzed by immunoblot with similar reduction in ECV proteins from *Fgf2* -/- stroma (*Figure 6E*).

## Fgf2 -/- stroma produces fewer exosomes and is less protective of BCR-ABL leukemia

To test the role of stromal *Fgf2* in an in vivo leukemia model, bone marrow from *Fgf2* +/+ mice was retrovirally transfected with BCR-ABL containing GFP as a marker (*Traer et al., 2012*) and used to transplant lethally irradiated FGF2 +/+ and -/- mice. This induces a very aggressive disease in mice that is more akin to AML than CML, and TKIs are only effective for a limited duration. Mice were treated with the ABL inhibitor nilotinib 75 mg/kg/day by oral gavage starting on day 14 post-transplant. Mice that were found to have aplastic marrow (unsuccessful transplantation) were excluded from analysis since their death was not related to leukemia (four mice in the *Fgf2* +/+ untreated group, one mouse in the *Fgf2* -/- untreated group, two mice in the *Fgf2* +/+ nilotinib group, and two mice in the *Fgf2* -/- nilotinib group). The survival curves of the remaining mice are shown in *Figure 7A*. The cohorts of untreated *Fgf2* +/+ and -/- both died rapidly from disease, as expected.

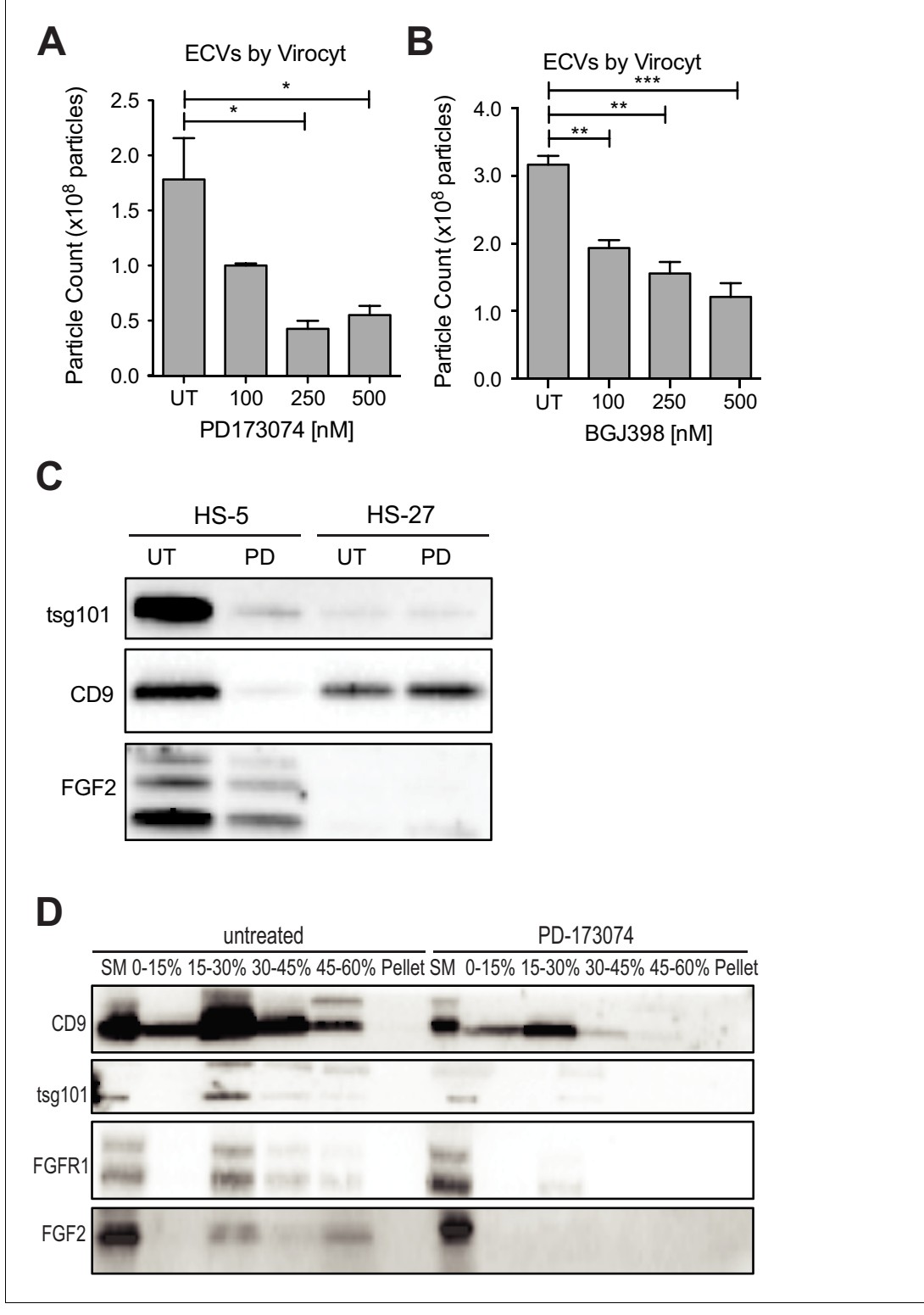

**Figure 5.** FGFR inhibition decreases exosome production in FGF2-expressing stroma. HS-5 cells were exposed to a gradient of the FGFR inhibitors (**A**) PD173074 and (**B**) BGJ-398 for 48 hr prior to collecting CM. ECVs were pelleted by ultracentrifugation at 100,000 g and quantified by Virocyt Virus Counter. Error bars indicate standard deviation and p values are indicated by *<0.05. (**C**) HS-5 and HS-27 cells were incubated in media ± 1 μM PD173074 for 72 hr prior to collecting ECVs. ECVs were analyzed by immunoblot for FGF2. The exosome markers CD9 and tsg101 are also shown. (**D**) HS-5 cells were plated in media ± 1 μM PD173074 for 72 hr. P100 fractions

*Figure 5 continued on next page*

*Figure 5 continued*

were obtained by ultracentrifugation, and further fractionated on a sucrose step-gradient. The interfaces (0–7.5%, 7.5–15%, 15–30%, 30–45%, and 45%-pellet) were collected, lysed and processed by immunoblot with antibodies against the exosomal markers CD9 and tsg101 as well as FGFR1 and FGF2.

DOI: https://doi.org/10.7554/eLife.40033.014

The following figure supplements are available for figure 5:

**Figure supplement 1.** FGFR inhibition reduces HS-5 cell exosome secretion.

DOI: https://doi.org/10.7554/eLife.40033.015

**Figure supplement 2.** Scanning electron microscopy of HS-5 cells shows altered membrane dynamics after FGFR inhibition.

DOI: https://doi.org/10.7554/eLife.40033.016

Nilotinib significantly increased survival of *Fgf2* +/+ and -/- mice compared to untreated mice, but the survival of the nilotinib-treated *Fgf2* -/- was also significantly longer than their *Fgf2* +/+ counter-rparts. To ensure equal engraftment of disease in both backgrounds, the blood and bone marrow was analyzed for GFP and found to be similar in both *Fgf2* +/+ and -/- mice at time of death (*Figure 7B* and *Figure 7—figure supplement 1*), suggesting that nilotinib was more effective at attenuating disease progression of BCR-ABL leukemia cells in an *Fgf2* -/- microenvironment. To directly evaluate the protective effect of ECVs on leukemia progenitor cells, ECVs were isolated from equal numbers of +/+ and -/- primary marrow stromal cells cultured with and without PD173074 treatment. Then, bone marrow from +/+ mice was retrovirally transfected with BCR-ABL and incubated with the ECVs overnight. The cells were washed, plated in methylcellulose with and without imatinib, and colonies counted after 8 days. Imatinib significantly reduced colony formation without ECVs, but ECVs from +/+ stroma almost completely reversed the inhibitory effects of imati-nib (*Figure 7C*). ECVs from PD173074-treated +/+ stroma or -/- stroma were not as protective, sug-gesting that *Fgf2* +/+ stroma more effectively protects BCR-ABL leukemia cells from the effects of kinase inhibition through secretion of protective exosomes. We confirmed the presence of *Fgf2* in microvesicles isolated from cultured *Fgf2* +/+ mouse stroma (*Figure 6*). To confirm that ECVs can be endocytosed by primary cells, lineage-negative hematopoietic progenitor cells were isolated from *Fgf2* +/+ mice and stained with a green lipophilic tracer (DiO) and incubated with ECVs from *Fgf2* +/+ or *Fgf2* -/- stromal cells stained with a red lipophilic tracer (DiI). Confocal microscopy confirmed internalization of fluorescently labeled primary stromal ECVs by murine progenitor cells (*Figure 7D* and *Figure 7—video 1* and *2*).

## Discussion

The normal hematopoietic microenvironment is altered by leukemia, and can protect leukemia cells from the effects of both chemotherapy and targeted kinase inhibitors (*Yang et al., 2014*; *Parmar et al., 2011*; *Manshouri et al., 2011*; *Traer et al., 2014*; *Traer et al., 2016*). Until recently, stromal protection of leukemia cells was thought to be largely mediated by secreted cytokines or through direct contact (review [*Meads et al., 2009*]). Here, we show that exosomes from bone mar-row stromal cells are transferred to leukemia cells, and protect them from kinase inhibitors. Exo-somes have previously been identified as important mediators of malignancy, including recent reports of leukemia exosomes modulating marrow stroma (*Paggetti et al., 2015*; *Huan et al., 2015*; *Peinado et al., 2012*; *Filipazzi et al., 2012*). We found that the reciprocal transfer also occurs, and that marrow stromal exosomes efficiently protect leukemia cells from targeted kinase inhibitors. Along with recent reports that entire mitochondria are transferred between stromal cells and leuke-mia cells during therapy (*Marlein et al., 2017*; *Moschoi et al., 2016*), our data adds to an increas-ingly complex and intimate relationship between marrow stromal cells and leukemia cells. Indeed, it is almost hard to imagine the leukemia cell in the niche as a separate entity given the direct exchange of organelles, ECVs, cell-cell signaling, and secreted cytokine signaling between stromal and leukemia cells. A better understanding of this relationship is important to develop better ways to eradicate leukemia cells and cure more patients.

Isolated reports have previously suggested that FGF2 is contained in ECVs (*Proia et al., 2008*; *Choi et al., 2016*), but FGF2 has also been reported to self-assemble into a pore-like structure on

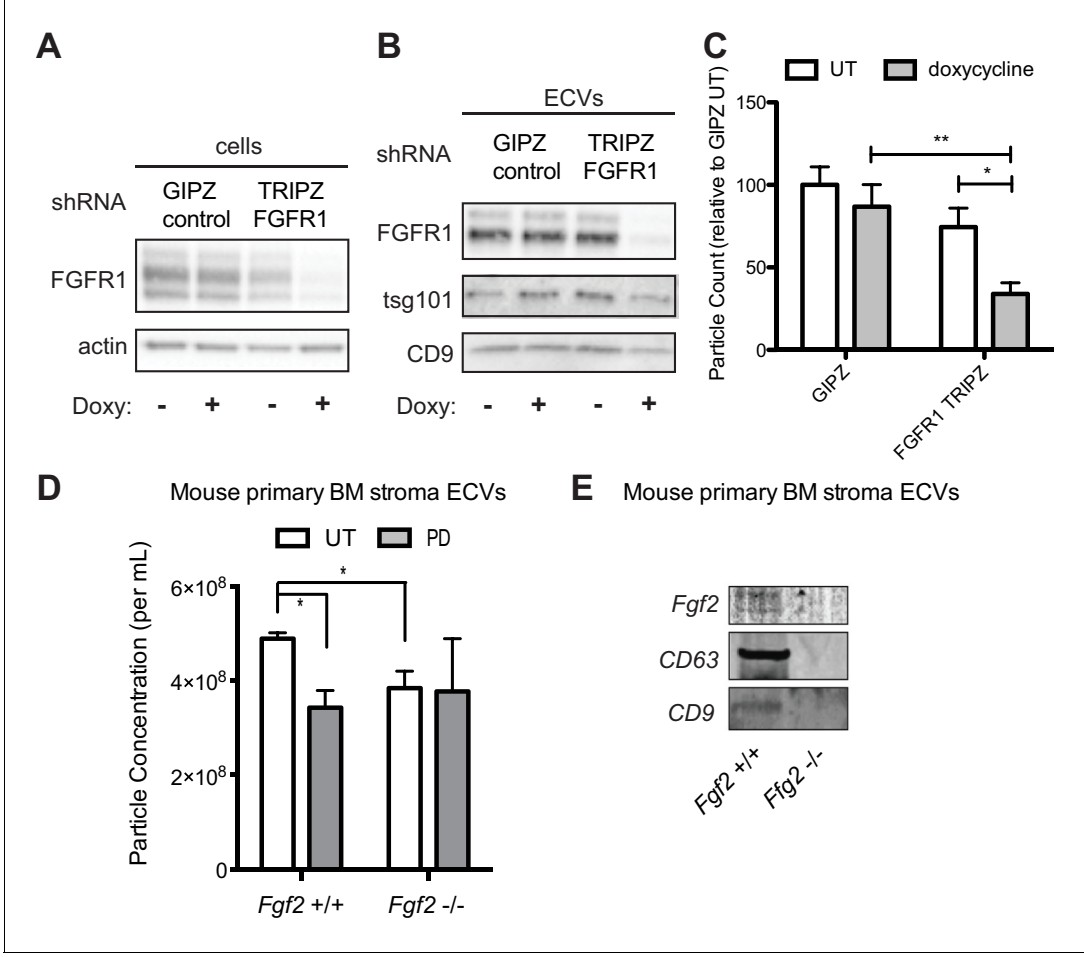

**Figure 6.** Genetic silencing of FGFR1 or deletion of FGF2 attenuates exosome secretion. A doxycycline-inducible lentiviral shRNA targeting FGFR1 was used to create a stable HS-5 cell line. The cells were then treated with doxycycline to induce FGFR1 silencing and compared to a GIPZ lentiviral control. (A) Silencing of FGFR1 expression is shown by immunoblot of cell lysates. ECVs from doxycycline-treated cells were analyzed by (B) immunoblot or (C) Virocyt Virus Counter. *p<0.05. (D) Bone marrow was isolated from *Fgf2* +/+ and -/- mice and cultured ex vivo to grow adherent marrow stroma. Equal numbers of cells were then plated, CM collected for 72 hr, and then ultracentrifuged to collect ECVs. The ECVs were quantified by Virocyt. *p<0.05. (E) Equal number of cultured marrow cells from *Fgf2* +/+ and -/- mice were plated and then ECVs collected by ultracentrifugation and analyzed by immunoblot.

DOI: https://doi.org/10.7554/eLife.40033.017

The following figure supplements are available for figure 6:

**Figure supplement 1.** Genetic silencing of FGFR1 by siRNA reduces exosome secretion and protection capacity of HS-5 stromal cells.
DOI: https://doi.org/10.7554/eLife.40033.018

**Figure supplement 2.** Genetic silencing of FGFR1 by CRISP/CAS9 reduces exosome secretion and protection capacity of HS-5 stromal cells.
DOI: https://doi.org/10.7554/eLife.40033.019

the cell membrane and mediate its own translocation with the help of extracellular heparan sulfate (*Steringer et al., 2015*; *Schäfer et al., 2004*). Compared to other soluble secreted cytokines, FGF2 was uniquely enriched in ECVs and exosomes (*Figure 2*), suggesting that secretion in ECVs is the primary mechanism of FGF2 paracrine signaling from marrow stromal cells. Since FGFR1 is also found on exosomes (*Figure 3D*), the FGF2-FGFR1 interaction on exosomes may play a direct role in loading FGF2 in exosomes and/or regulate secretion. FGFR inhibitors also increase the amount of FGFR1 protein in stromal cells as measured by immunoblot, consistent with a role in receptor cycling and/or reduced secretion in exosomes.

Similar to our observations, epidermal growth factor receptor has been shown to be secreted on ECVs, and secretion is increased after ligand stimulation (*Sanderson et al., 2008*; *Perez-*

*Torres et al., 2008*). Likewise, overexpression of oncogenic HER2 in breast cancer cell lines resulted in qualitative differences in microvesicle content (*Amorim et al., 2014*), suggesting a role for activated receptor tyrosine kinases in exosome production and secretion. Receptor-mediated endocytosis is the first step of exosome biogenesis (*Théry et al., 2002*), suggesting that inhibitors of receptor tyrosine kinases may act at this step. How FGFR1 is positioned in the exosome membrane (inside or out), how FGF2 binds FGFR1 in exosomes, and how exosomal FGF2 activates FGFR1 in leukemia cells, are areas of active investigation.

FGF2 has been previously implicated in hematologic malignancy progression and development of resistance (*Sato et al., 2002*; *Chesi et al., 2001*). Elevated levels of FGF2 have previously been measured in the serum of CML and AML patients (*Aguayo et al., 2000*; *Aguayo et al., 2002*), as well as in the bone marrow of AML patients, where it was reported to function as an autocrine promotor of proliferation (*Bieker et al., 2003*). We found that FGF2 expression was increased in CML and AML stroma during the development of resistance to kinase inhibitors, indicating that FGF2 expression is a regulated autocrine growth factor for stroma (*Traer et al., 2016*). This is consistent with the role of FGF2-FGFR1 autocrine expansion of stroma in stress-induced hematopoiesis (*Itkin et al., 2012*; *Zhao et al., 2012*) and suggests that leukemia cells are able to hijack the FGF2 stress response for survival. The regulation of FGF2-FGFR1 signaling is also supported by the positive correlation in expression of both FGF2 and FGFR1 in a subset of primary AML marrow samples (*Figure 4G*), indicating that this pathway can be selectively activated. FGFR inhibitors not only inhibit autocrine growth of stroma, but reduce exosome secretion and significantly alter the protective ability of stromal cells (*Figures 4A* and *7*). Since exosomes contain a complex mixture of proteins, cytokines, lipids and microRNAs (all of which potentially contribute to leukemia cell protection), inhibiting secretion of exosomes is a promising approach to blunting this complex mechanism of resistance.

In summary, FGF2 is a regulated autocrine growth factor for marrow stroma that is important in reprogramming the marrow stroma during development of resistance to TKIs. FGF2-FGFR1 activation in marrow stroma leads to increased secretion of exosomes, which are protective of leukemia cells in both in vitro and in vivo models. Given the inevitable development of clinical resistance to TKIs (FLT3 ITD AML in particular), addition of FGFR inhibitors to directly modulate the leukemia niche is a promising approach to improve the durability of response.

## Materials and methods

**Key resources table**

| Reagent type (species) or resource | Designation | Source or reference | Identifiers | Additional information |
|---|---|---|---|---|
| Gene (homo sapiens) | FGF2 | NA | | |
| Gene (mus musculus) | *Fgf2* | NA | | |
| Gene (homo sapiens) | FGFR1 | NA | | |
| Gene (mus musculus) | *Fgfr1* | NA | | |
| Strain, strain background (mus musculus) | *Fgf2*tm1Doe/J *Fgf2* +/+ and -/- mice | Jackson Laboratory | RRID:MGI:2679603 | |
| Genetic reagent (homo sapiens) | FGF2 | Thermo Fisher Scientific | | shRNA in TRIPZ lentiviral vector |
| Genetic reagent (homo sapiens) | FGFR1 | Thermo Fisher Scientific | | shRNA in TRIPZ lentiviral vector |
| Genetic reagent (homo sapiens) | | AddGene | | GeCKO lentiCRISPRv2 hSpCas9 and guide RNA |
| Genetic reagent (homo sapiens) | FGF2-1 | GenScript | | CRISPR/Cas nine guide RNA design |

*Continued on next page*

*Continued*

| Reagent type (species) or resource | Designation | Source or reference | Identifiers | Additional information |
|---|---|---|---|---|
| Genetic reagent (homo sapiens) | FGF2-2 | GenScript | | CRISPR/Cas nine guide RNA design |
| Genetic reagent (homo sapiens) | FGFR1-1 | GenScript | | CRISPR/Cas nine guide RNA design |
| Genetic reagent (homo sapiens) | FGFR1-2 | GenScript | | CRISPR/Cas nine guide RNA design |
| Genetic reagent (mus musculus) | pMIG with BCR-ABL and GFP | | | murine retrovirus |
| Cell line (homo sapiens) | MOLM14 | Dr. Yoshinobu Matsuo | RRID:CVCL_7916 | |
| Cell line (homo sapiens) | K562 | American Type Culture Collection | RRID:CVCL_0004 | |
| Cell line (homo sapiens) | HS-5 | Dr. Beverly Torok-Storb | RRID:CVCL_3720 | |
| Cell line (homo sapiens) | HS-27 | Dr. Beverly Torok-Storb | RRID:CVCL_0335 | |
| Antibody | Mouse monoclonal anti-FGFR1 | Cell Signaling | 9740 | Dilution 1:1000 |
| Antibody | Rabbit polyclonal anti-FGF2 | Santa Cruz | Sc-79 | Dilution 1:500 |
| Antibody | Rabbit monoclonal anti-CD63 | ABCAM | ab134045 | Dilution 1:1000 |
| Antibody | Rabbit polyclonal anti-CD9 | Santa Cruz | Sc-9148 | Dilution 1:200 |
| Antibody | Mouse monoclonal anti-tsg-101 | Santa Cruz | Sc-7964 | Dilution 1:200 |
| Antibody | Mouse monoclonal anti-actin | Millipore | MAB1501 | Dilution 1:5000 |
| Peptide, recombinant protein | FGF2 (human) | Peprotech | | |
| Commercial assay or kit | Thermo Scientific lentiviral transfection kit | | | |
| Chemical compound, drug | quizartinib (AC220) | LC labs | | |
| Chemical compound, drug | imatinib | LC labs | | |
| Chemical compound, drug | nilotinib | SelleckChem | | |
| Chemical compound, drug | PD173074 | SelleckChem | | |
| Chemical compound, drug | BGJ-398 | SelleckChem | | |
| Chemical compound, drug | doxycycline | Fisher | | |
| Software, algorithm | CellProfiler | Cell area | | |

## Cell lines

The human cell line MOLM14 was generously provided by Dr. Yoshinobu Matsuo (Fujisaki Cell Center, Hayashibara Biochemical Labs, Okayama, Japan). The human cell line K562 was obtained from the American Type Culture Collection (Manassas, VA, USA). The human stromal cell lines HS-5 and HS-27a were kindly provided by Dr. Beverly Torok-Storb (Fred Hutchinson Cancer Research Center, Seattle, WA). Cells were maintained in RPMI1640 media supplemented with 10% fetal bovine serum

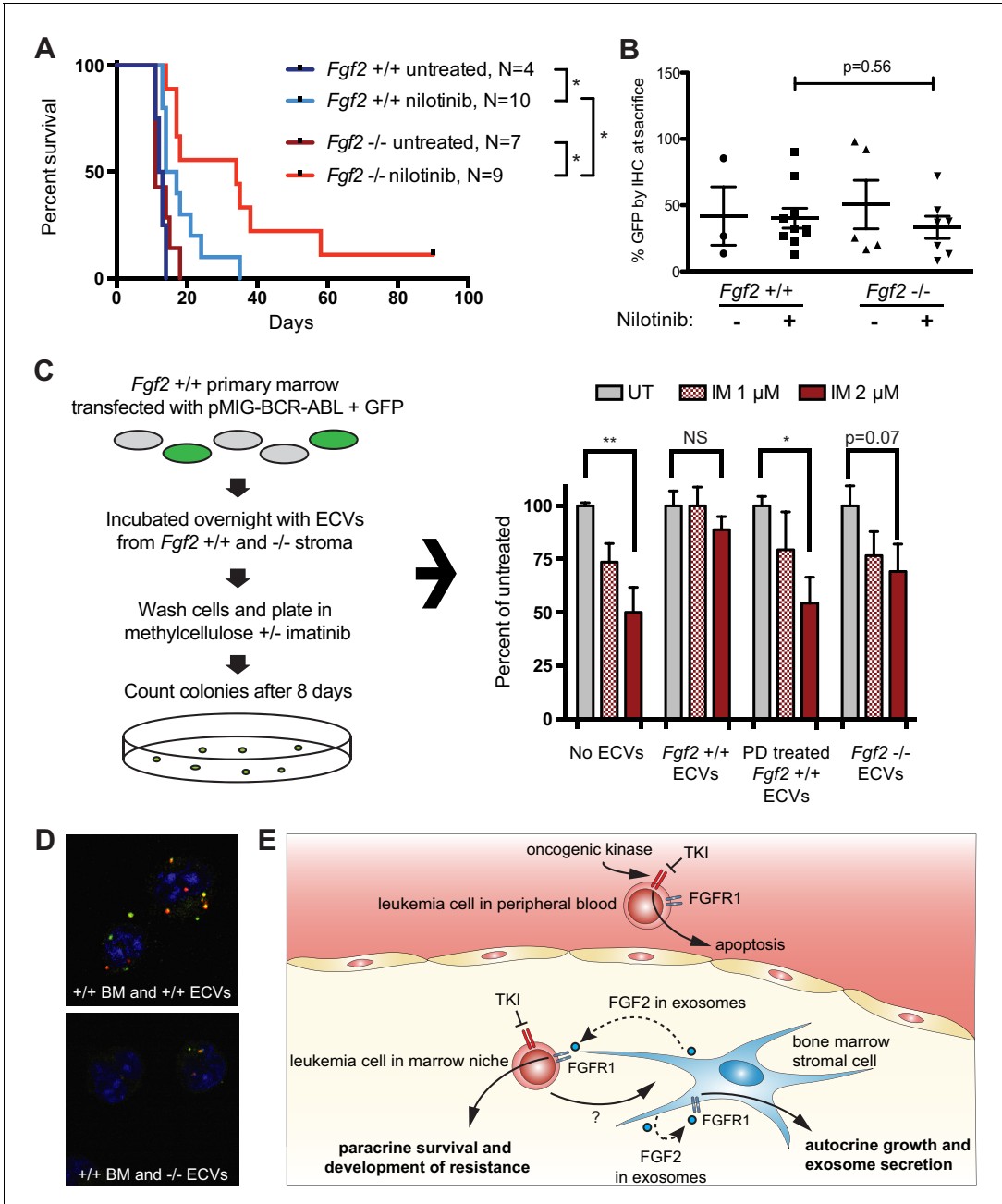

**Figure 7.** *Fgf2* -/- mice survive significantly longer with TKI therapy in a murine BCR-ABL leukemia model. *Fgf2* +/+ bone marrow was removed from donor mice and spinoculated with pMIG BCR-ABL retrovirus containing an IRES-GFP marker. The transfected bone marrow was then transplanted into lethally irradiated *Fgf2* +/+ or -/- recipients. Mice were treated with 75 mg/kg/day nilotinib by oral gavage starting on day 11 of transplant. (**A**) Survival curves of untreated and nilotinib-treated *Fgf2* +/+ and -/- mice. (**B**) GFP in peripheral blood was evaluated weekly and at time of euthanasia to quantify disease burden. The average GFP (percent of nucleated cells) is shown and did not differ significantly between groups indicating that all animals developed similar disease burden. Error bars indicate standard deviation. (**C**) Bone marrow cells from *Fgf2* +/+ mice were spinoculated with pMIG BCR-ABL retrovirus containing GFP-IRES. The cells were then incubated with ECVs obtained from *Fgf2* +/+ and -/- primary stroma cultured alone or with 500 nM PD173074. The next day the incubated cells were washed three times to remove cytokines and exosomes and plated in cytokine-free methylcellulose ± imatinib. After 8 days, colonies were counted and normalized to untreated condition. Graph shown on right. Error bars indicate standard error of the mean. *p<0.05 and **p<0.005. (**D**) Lineage-negative bone marrow cells were isolated from *Fgf2* +/+ mice and cells were stained with DiO (green) tracer, washed, and immobilized on Poly-D-lysine coated chamber slides. ECVs from bone marrow stroma of *Fgf2* +/+ or -/- mice were stained with DiI (red) tracer and added to the cells for a 24 hr incubation. Slides were stained with DAPI (blue) and imaged by confocal fluorescent microscopy. Movie of the z-stack images are included as *Figure 7—video 1* and *2*. (**E**) Model of bone marrow stromal FGF2 autocrine signaling and paracrine protection of leukemia cells by FGF2-containing exosomes.

*Figure 7 continued on next page*

*Figure 7 continued*

DOI: https://doi.org/10.7554/eLife.40033.020

The following video and figure supplement are available for figure 7:

**Figure supplement 1.** *Fgf2* +/+ and -/- mice demonstrate engraftment of leukemia by immunohistochemistry (IHC) of GFP.

DOI: https://doi.org/10.7554/eLife.40033.021

**Figure 7—video 1.** ECVs collected from *Fgf2* +/+ primary marrow stromal cell cultures were stained with DiI (red) and lineage-depleted hematopoietic cells from *Fgf2* +/+ marrow were stained with DiO (green) and incubated for 24 hr at 37°C as described in Materials and methods.

DOI: https://doi.org/10.7554/eLife.40033.022

**Figure 7—video 2.** ECVs collected from *Fgf2* -/- primary marrow stromal cell cultures were stained with DiI (red) and lineage-depleted hematopoietic cells from *Fgf2* +/+ marrow were stained with DiO (green) and incubated for 24 hr at 37°C as described in Materials and methods.

DOI: https://doi.org/10.7554/eLife.40033.023

(FBS), 100 U/ml penicillin/100 µg/ml streptomycin, 2 mM L-glutamine, and 0.25 µg/ml fungizone (referred to as **R10**) at 37°C in 5% $CO_2$. Exosome-depleted FBS was pre-cleared by ultracentrifugation at 100,000 g for 2 hr at 4°C. Cell lines were validated by genetic and functional analysis based upon previous reported characteristics. Cell lines were tested monthly for mycoplasma infection and discarded if found to be infected.

## ECV isolation

HS-5 cells grown to 90–100% confluence in 15 cm dishes were washed in 8 ml PBS, and incubated in 12 ml exosome-depleted R10 overnight. The media was collected, cleared of debris (2 × 2000 g spin, 10 min), and ultracentrifuged at 100,000 g for 2 hr at 4°C. The resulting supernatant (S100) was poured off, and 100 ul PBS was added to the ECV pellet (P100). This was shaken for 4 hr at 4°C at 2000 rpm. P100 was used fresh or stored at −80°C with 10% DMSO.

## Sucrose density step-gradient

Layers of sucrose (60%, 45%, 30%, 15%, 7.5%, and 0%) were carefully pipetted into ultracentrifuge tubes. ECVs were added on top, and the tube ultracentrifuged at 100,000 g for 90 min at 4°C. The sucrose interfaces (45–60, 30–45, 15–30, 7.5–15, 0–7.5) were collected with a micropipette, washed in PBS, and pelleted at 100,000 g for 2 hr at 4°C.

## ECV quantitation

ECVs were quantified by Nanosight LM10 or by Virocyt Virus Counter 3100, following manufacturers' protocols.

## Inhibitors and cytokines

Quizartinib (AC220) was purchased from LC labs (Woburn, MA, USA). Nilotinib, PD173074 and BGJ-398 were purchased from SelleckChem (Houston, TX, USA). Imatinib was purchased from LC labs (Woburn, MA, USA). Recombinant FGF2 was purchased from Peprotech (Rocky Hill, NJ, USA).

## Viability assays

Viability was assessed with MTS reagent, CellTiter 96 AQueous One Solution Proliferation Assay from Promega Corporation (Madison, WI, USA) or by Guava ViaCount flow cytometer assay (Millipore, Burlington, MA, USA).

## Immunoblot analysis

Treated cells were washed in PBS before adding lysis buffer (Cell Signaling, Danvers, MA, USA) supplemented with Complete protease inhibitor (Roche, Indianapolis, IN, USA) and phosphatase inhibitor cocktail-2 (Sigma-Aldrich, St. Louis, MO, USA). Proteins were fractionated on 4–15% Tris-glycine polyacrylamide gels (Criterion gels, Bio-Rad), transferred to PVDF membranes, and probed with antibodies: FGFR1, fibronectin (Cell Signaling, Danvers, MA, USA); CD9, FGF2, calreticulin, tsg101 (Santa Cruz Biotechnology, Dallas, TX, USA), CD63 (Abcam, Boston, MA, USA), and actin (MAB1501, Millipore, Burlington, MA, USA).

## Stromal cell cytokine ELISA

Stromal CM, S100 and ECVs were lysed with 0.1% NP-40 for 30 min, centrifuged 3,000 rpm for 10 mins, and 50 µl supernatant was incubated with the magnetic beads overnight and assayed as per manufacturer's instructions (Luminex Multiplex magnetic beads 30-plex Assay, Life Technologies).

## Primary bone marrow stromal cultures

Bone marrow aspirates were obtained from AML patients after informed consent under the OHSU Institutional Research Board protocol IRB0004422, and were processed as previously described (*Viola et al., 2016*). After Ficoll, the red cell pellets were incubated with ACK for 30 min on ice to lyse red cells, and plated on 15 cm dishes in MEM-α supplemented with 20% fetal bovine serum (FBS), 100 U/ml penicillin/100 µg/ml streptomycin, 2 mM L-glutamine, and 0.25 µg/ml fungizone at 37°C in 5% $CO_2$. After 48 hr, non-adherent cells were removed and new media was added. This step was repeated after an additional 24 hr. Cells were then incubated for 1–3 weeks with media changes every 7 days, until patchy proliferation became apparent. Cells were trypsinized and replated to facilitate homogenous growth. Cells were expanded over a maximum of 3 passages before use in experiments. Murine primary stroma was isolated from harvested femur marrow without ACK treatment and then cultured as above. Primary stromal samples were analyzed after >2 weeks growth.

## Transmission electron microscopy

Stromal cell exosomes were isolated by sucrose step-gradient then washed in 0.22 µm filtered PBS. 10 µl was deposited onto glow discharged carbon formvar 400 mesh copper grids (Ted Pella 01822 F) for 3 min, rinsed 15 secs in water, wicked on Whatman filter paper 1, stained for 45 secs in filtered 1.33% (w/v) uranyl acetate, wicked and air dried. Samples were imaged at 120kV on a FEI Tecnai Spirit TEM system. Images were acquired as 2048 × 2048 pixel, 16-bit gray scale files using the FEI's TEM Imaging and Analysis (TIA) interface on an Eagle 2K CCD multiscan camera.

## Fluorescent confocal microscopy

MOLM14 and K562 cells were stained with DiO (Thermo Fisher) according to manufacturer's protocol. HS-5 ECVs were stained with DiI (Thermo Fisher), washed with PBS, and collected by ultracentrifugation. For experiments using mouse bone marrow, cells were isolated from femurs and tibias, RBCs were lysed using ACK buffer (0.8% NH4Cl and 0.1 mMEDTA in KHCO3 buffer; pH 7.2–7.6), and lineage-negative cells were isolated by MACS cell separation with the human lineage cell depletion kit (Milteny Biotec). Cells were incubated with a cytokine mix (IL-3, IL-6, SCF) in addition to DiO. DiO-stained cells were combined with DiI-stained ECVs and incubated for 24 hr at 37°C. Cells were washed, placed on poly-D-lysine coated chamber slides, and DAPI-stained. Z-stack imaging was performed on an Olympus IX71 inverted microscope. Images were processes using the Fiji software package (*Schindelin et al., 2012*).

## Proteinase K digestion

ECVs, or exosomes isolated by sucrose step-gradient, were resuspended in proteinase K buffer (Tris-HCl pH8, 10 mM CaCl2) and then incubated with 200 µg/ml proteinase K at room temp for 30 min. 5 µL 0.1 M PMSF and SDS loading buffer was added and samples were incubated at 98°C for 5 min to stop reaction prior to immunoblots.

## Cell morphology analysis

HS-5 and HS-27 cells were grown to 90% confluence in 4-well chamber microscope slides in R10 ±250 nM PD173074. Cells were stained with lipophilic tracer DiI, washed, and stained with DAPI. Cells were imaged with Zeiss Axio Observer fluorescent microscope at 10X using AxioVision software. Images were uploaded into CellProfiler software and analyzed for cell size. Cell diameter was determined as $diameter\ [\mu m] = \sqrt{pixels \times 0.394\ \mu m^2/pixel}$.

## shRNA

TRIPZ inducible lentiviral FGF2 and FGFR1 shRNA were purchased from Thermo Fisher Scientific Dharmacon RNAi Technologies (Waltham, MA, USA), along with Dharmacon's trans-lentiviral shRNA packaging kit with calcium phosphate transfection reagent and HEK293T cells. HS-5 and HS-27 cells

were transfected with GIPZ control or FGFR1 TRIPZ, per manufacturer's protocol. TurboRFP/shRNA expression was induced with 1 µg/ml doxycycline (Fisher) for 48 hr, cells were washed in PBS, and then media replaced with exosome-depleted R10 +1 µg/ml doxycycline. Cells and CM were collected after 72 hr for analysis.

### CRISPR/Cas9 targeted genome editing

The vector GeCKO lentiCRISPRv2 was obtained from Addgene. This plasmid contains two expression cassettes, hSpCas9 and the chimeric guide RNA. Guide RNA sequences were obtained from GenScript (*Sanjana et al., 2014*), and oligos with 5' overhang for cloning into lentiCRISPRv2 were manufactured by Fisher Scientific. The vector was digested with BsmBI and dephosphorylated, the plasmid was gel-purified, and oligonucleotides were ligated after annealing and phosphorylation. Plasmid was amplified in Stbl3 bacteria, purified, and lentivirus was generated in HEK293T cells. Transduced HS-5 cells were selected in puromycin for 5 days, and cultured an additional 5 days before assessing knockout.

### Murine BCR-ABL leukemia experiments

Animal studies were approved by the OHSU Institutional Animal Care and Use Committee. *Fgf2*$^{tm1Doe}$/J were purchased from Jackson Laboratory to breed homozygous +/+ and -/- littermates. Bone marrow from 5-FU treated *Fgf2* +/+ donors was spinoculated with pMIG containing BCR-ABL and IRES-GFP reporter as previously described (*Traer et al., 2012*; *Agarwal et al., 2008*) and 2 × 10$^6$ cells were retro-orbitally injected into lethally irradiated (2 × 450 cGy administered 4 hr apart) *Fgf2* +/+ and -/- recipients. 75 mg/kg/day nilotinib was administered by oral gavage and mice were monitored weekly with cell blood counts and FACS analysis of GFP in peripheral blood. Diseased mice were subjected to detailed histopathologic analysis. For colony assays, ECVs were isolated (as above) from equal numbers of *Fgf2* +/+ and -/- primary stromal cells cultured on 10 cm plates for 3 days, with and without 500 nM PD173074 (3 day pre-treatment and 3 days during ECV collection). Bone marrow from FGF2 +/+ mice was spinoculated with pMIG containing BCR-ABL and IRES-GFP reporter as above, incubated with ECVs overnight and washed 3x the next day. 3% of cells were GFP positive by FACS, and 4 × 10$^3$ cells were then plated in 1 ml of MethoCult M3234 Methylcellulose Medium for Mouse Cells without cytokines (Stemcell Technologies) in triplicate. Mouse bone marrow colonies larger than 50 cells were counted after 8 days.

### Statistical methods

Graphical and statistical data were generated with Microsoft Excel or GraphPad Prism (GraphPad Software, La Jolla, CA, USA). *P* value < 0.05 was considered significant.

### Conflict of interest disclosure

The authors declare no competing interests. Dr. Druker is currently principal investigator or co-investigator on Novartis clinical trials. His institution, OHSU, has contracts with these companies to pay for patient costs, nurse and data manager salaries, and institutional overhead. He does not derive salary, nor does his lab receive funds from these contracts.

## Acknowledgements

We would like to thank Cristina Tognon and Kara Johnson for administrative assistance, and Angie Rofelty for assistance with primary bone marrow stromal cultures. Electron microscopy was performed at the Multiscale Microscopy Core (MMC) with technical support from the Oregon Health and Science University (OHSU)-FEI Living Lab and the OHSU Center for Spatial Systems Biomedicine (OCSSB).

## Additional information

### Competing interests

Brian J Druker: Is currently principal investigator or co-investigator on Novartis clinical trials. His institution, OHSU, has contracts with these companies to pay for patient costs, nurse and data manager salaries, and institutional overhead. He does not derive salary, nor does his lab receive funds from these contracts. The other authors declare that no competing interests exist.

### Funding

| Funder | Grant reference number | Author |
|---|---|---|
| American Cancer Society | MRSG-17-040-1 - LIB | Elie Traer |
| Howard Hughes Medical Institute | | Brian J Druker |
| National Institutes of Health | 5F30CA186477-03 | Javidi-Sharifi Nathalie |

The funders had no role in study design, data collection and interpretation, or the decision to submit the work for publication.

### Author contributions

Nathalie Javidi-Sharifi, Conceptualization, Data curation, Formal analysis, Funding acquisition, Investigation, Visualization, Methodology, Writing—original draft, Writing—review and editing; Jacqueline Martinez, Data curation, Formal analysis, Validation, Investigation, Visualization, Methodology, Writing—original draft; Isabel English, Data curation, Formal analysis, Investigation, Methodology, Writing—original draft; Sunil K Joshi, Data curation, Formal analysis, Investigation, Visualization; Renata Scopim-Ribeiro, Data curation, Investigation, Visualization; Shelton K Viola, Data curation, Investigation; David K Edwards V, Anupriya Agarwal, Claudia Lopez, Danielle Jorgens, Resources, Investigation; Jeffrey W Tyner, Resources, Writing—review and editing; Brian J Druker, Resources, Supervision, Writing—review and editing; Elie Traer, Conceptualization, Data curation, Formal analysis, Supervision, Funding acquisition, Validation, Investigation, Visualization, Methodology, Writing—original draft, Project administration, Writing—review and editing

### Author ORCIDs

Elie Traer (iD) http://orcid.org/0000-0001-8844-2345

### Ethics

Human subjects: Bone marrow aspirates were obtained from AML patients after informed consent under the OHSU Institutional Research Board protocol IRB0004422.
Animal experimentation: Animal studies were carried out under approved OHSUInstitutional Animal Care and Use Committee, Protocol IP00000723. Number of animals for study and unnecessary suffering was minimized as much as possible.

### Decision letter and Author response

Decision letter https://doi.org/10.7554/eLife.40033.028
Author response https://doi.org/10.7554/eLife.40033.029

## Additional files

### Supplementary files

• Supplementary file 1. Additional information on antibodies used in paper.
DOI: https://doi.org/10.7554/eLife.40033.024

• Transparent reporting form
DOI: https://doi.org/10.7554/eLife.40033.025

## Data availability

All data generated or analysed during this study are included in the manuscript and supporting files. Movies for confocal images in Figures 1 and 7 is provided.

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
