## [Decision Letter]

[**Editorial note:** This article has been through an editorial process in which the authors decide how to respond to the issues raised during peer review. The Reviewing Editor's assessment is that all the issues have been addressed.]

Thank you for submitting your article "FGF2-*FGFR1* signaling regulates release of leukemia-protective exosomes from bone marrow stromal cells" for consideration by *eLife*. Your article has been reviewed by two peer reviewers, one of whom is a member of our Board of Reviewing Editors, and the evaluation has been overseen by Jeffrey Settleman as the Senior Editor. The following individual involved in review of your submission has agreed to reveal their identity: Thomas O'Hare (Reviewer #2).

The Reviewing Editor has highlighted the concerns that require revision and/or responses, and we have included the separate reviews below for your consideration. If you have any questions, please do not hesitate to contact us.

Summary:

This manuscript addresses the mechanism(s) by which protective signaling from the leukemia microenvironment may promote leukemia cell persistence, the development of drug resistance, and ultimately disease relapse. The authors claim that fibroblast growth factor 2 (FGF2) is released in exosomes from bone marrow stromal cells and that the subsequent endocytic entry of FGF2 into leukemia cells protects the leukemia cells from oncoprotein kinase targeted inhibitors. The authors claim that expression of FGF2, and its receptor *FGFR1*, is increased in a subset of stromal cell lines and primary AML stroma. Moreover, they claim that increased FGF2/*FGFR1* signaling is associated with increased exosome secretion such that pharmacologic or genetic inhibition of *FGFR1* signaling inhibition interrupts stromal autocrine growth and decreases secretion of FGF2-containing exosomes resulting in decreased protection of leukemia cells. Support for this hypothesis comes from the observation that, when BCR-ABL-driven leukemia cells are transplanted into FGF2-deficient mice, they display enhanced therapeutic benefit compared to FGF2-proficient mice. Hence, the authors claim that inhibition of the FGF2->*FGFR1* signaling axis may be a strategy to prevent or overcome the development of TKI resistance in leukemia. Overall, the findings of this study are supported by strong, well-controlled experimental data and the manuscript is clearly written. However, both reviewers have both major and minor concerns, which if addressed by the authors, will substantially improve the quality of the manuscript.

Separate reviews (please respond to each point):

*Reviewer #1:*

This manuscript address the mechanism(s) by which protective signaling from the leukemia microenvironment may promote leukemia cell persistence, the development of drug resistance, and ultimately disease relapse. The authors claim that fibroblast growth factor 2 (FGF2) is released in exosomes from bone marrow stromal cells and that the subsequent endocytic entry of FGF2 into leukemia cells protects the leukemia cells from oncoprotein kinase targeted inhibitors. The authors claim that expression of FGF2 and its receptor *FGFR1*, are increased in a subset of stromal cell lines and primary AML stroma. Moreover, they claim that increased FGF2/*FGFR1* signaling is associated with increased exosome secretion such that pharmacologic or genetic inhibition of *FGFR1* signaling inhibition interrupts stromal autocrine growth and decreases secretion of FGF2-containing exosomes resulting in decreased protection of leukemia cells. Support for this hypothesis comes from the observation that, when BCR-ABL driven leukemia cells are transplanted into FGF2 deficient mice, they display enhanced therapeutic benefit compared to FGF2 proficient mice. Hence, the authors claim that inhibition of the FGF2->*FGFR1* signaling axis may be a strategy to prevent or overcome the development of TKI resistance in leukemia.

The authors conclude from the data in Figure 1C that ECVs produced by HS-5 cells are internalized by endocytosis. What evidence do the authors have the ECVs are actually entering cells by endocytosis and, if entry is by endocytosis, is it clathrin-mediated endocytosis, caveolae, micropinocytosis or phagocytosis?

The data in Figures2E/F on protection of exosomal FGF2 from proteinase K digestion are quite equivocal in that the partial protection observed is not entirely consistent with the authors' interpretations. Indeed, the protection described in Figure 2F is so marginal as to make this reviewer wonder how reproducible the data are and whether, or not, a modest increase in incubation time with proteinase K would lead to complete proteolysis of FGF2. Indeed, the authors' conclusions that, "….. FGF2 is contained within exosomes but also likely associated with proteins on the surface of exosomes." is rather ambivalent.

In Figure 3, the authors convincingly demonstrate that HS-5 CM contains more exosomes that HS-27 CM. They also note that the receptor for FGF2, *FGFR1*, is also present in exosomes. One obvious question is whether *FGFR1* is tyrosine phosphorylated on sites that are stimulated by FGF2, which would suggest that a portion of FGF2 and *FGFR1* are in complex with one another.

In Figure 5, the authors show that TKIs of *FGFR1* signaling reduce exosome release from HS-5 stromal cells. This data are convincing, but it is also rather phenomenological and lacks a mechanism by which *FGFR1* might regulate the release of exosomes by bone marrow stromal cells. Also, the fact that this occurs in one cell line does not lend confidence that this is a generalizable conclusion.

In Figure 6, the use of Tet-regulated shRNA targeting shRNA is a good complement to the use of pharmacological TKIs of *FGFR1* signaling. However, the effects of shFGFR1 on exosomal FGF2 should be assessed in Figure 6B – since the effects of shFGFR1 on tsg101 and CD9 are barely discernible given the poor quality of immunoblots presented in this figure. Furthermore, the authors need to better explain why their CRISPR/CAS9 efforts to silence the expression of either *FGFR1* or FGF2 resulted in partial loss of *FGFR1* or FGF2 expression. Moreover, whereas partial silencing of *FGFR1* had almost no discernible effect on exosomal tsg101 or CD9, partial silencing of FGF2 had an appreciable effect on exosomal tsg101 and CD9. These data are somewhat discordant and are not adequately explained in the text.

The experiments presented in Figure 7 were nicely conceived and elegantly executed. This figure makes a case for an important role of stromal derived, exosome delivered FGF2 as a relevant mechanism of TKI resistance in leukemia.

*Reviewer #2:*

The findings of the study are supported by strong, well-controlled experimental data. The paper is clearly written. My major concerns are the heavy reliance on HS-5 as proxy for the leukemic microenvironment and that the story is almost entirely presented from the perspective of the stroma. Several mechanistic questions pertaining to the leukemic cells are asked below, with the goal of helping the reader to better understand how the authors are viewing and rationalizing these points.

1) A stated theme of the study is that the normal hematopoietic microenvironment is altered by leukemia. While this is undoubtedly correct, the choice of CM from a cell line derived from normal bone marrow would seem to limit applicability and relevance to the leukemic microenvironment.

2) Why doesn't the presence of an FGFR inhibitor (PD173074) in experiments in which CM is provided to leukemic cell lines dampen the effectiveness of HS-5 CM? The authors make a convincing case that FGF2-laden exosomes are taken up by the leukemic cell lines and that this provides a protective effect against TKIs. They also show that FGFR inhibition (or [partial] *FGFR1* deletion) in HS-5 stromal cells decreases exosome production/secretion and probably alters the content of exosomes. However, the story from the point of view of the leukemic cell lines is largely unexplored and mechanistically vague. Does *FGFR1* deletion in the leukemic cell lines change the effectiveness of HS-5 CM? Does PD173074 completely shut off *FGFR1* signaling in the leukemic cell lines? Perhaps I am missing something, but I find it surprising that inhibition of *FGFR1* (in the leukemic cells) does not at least partially reverse the protective effects of HS-5 CM. This is largely based on the authors' reasonable assertion that the most important exosome cargo is FGF2.

3) Several TKIs used in CML have target profiles that include FGFR, most notably the pan-FGFR TKI, ponatinib. Midostaurin, the currently approved frontline TKI for FLT3-mutated AML, inhibits *FGFR1* and 2. Would patients treated with these TKIs derive additional benefit from collateral inhibition of *FGFR1* in stromal cells (and possibly in leukemic cells)? I expected to see a comparison of imatinib and ponatinib in the study. I also thought this would be worth testing in the mouse model, as mentioned below.

4) Panels 4F,G show data for primary AML stromal cells (n = 42). The legend should specify how many AML patients are represented and how many data points are from each patient. Are some of these FLT3-ITD patients, and if so, can you indicate them as colored dots? This display item and the corresponding Materials and methods, Results and Discussion require more detail. This has potential to be an intriguing part of the story but the presentation is murky. Is it possible to include a comparison to HS-5 and HS-27 in these panels? Are the *FGFR1* and FGF2 levels (and their correlations) substantially different from normal bone marrow? Should we expect them to be? Please consider strengthening this part of the manuscript. I could not glean much as presented.

Minor Comments:

1) Figure 1: In panel A, what is the result if the same experiment is carried out with ponatinib (10-25 nM)? dasatinib (10-25 nM)? In panel B, with midostaurin (1 µM)? This comment is somewhat redundant with major point (3) above; here, I am requesting that the data are included, at least in the supplement.

2) Why was 1 µM imatinib used? This concentration, at least at the 48 h timepoint, does not reduce proliferation (Figure 1A) to the extent that AC220 does in MOLM14 cells. Thus, it is difficult to determine the extent to which CM provides a protective effect. What happens if a higher concentration of imatinib is used, or if a more potent TKI is used at a concentration that matches the AC220 magnitude of reduced proliferation?

3) Regarding the FGFR inhibitors:

– Why two? BGJ-398 is described as 'selective' but I did not find the rationale for needing it in the study (which began with PD173074).

– What is the direct evidence/readout to confirm that these inhibitors are working (and to measure the extent of inhibition)?

4) Perhaps it's not a reasonable comparison, but it is surprising that shRNA-based elimination of *FGFR1* does not reduce FGF2 levels in HS-5 cells (Figure 6A), but FGFR inhibition does (Figure 5C). Also, the HS-5 CRISPR *FGFR1*.1 and 1.2 lines show reduced FGF2. The HS-5 CRISPR FGF2.1 and 2.2 lines show enhanced *FGFR1* but reduced *FGFR1* in ECVs. Can the authors comment on / explain these points?

5) In Figure 6B, I would like to be able to evaluate whether reduced *FGFR1* brings about a reduction in FGF2, but this data is not provided. In general, both FGF2 and *FGFR1* should be shown in immunoblot stacks. Sometimes, it is one or the other.

6) Figure 7:

– why is nilotinib used in 7A? All other data are with imatinib.

– the main point of the paper seems to be that interrupting the *FGFR1*-FGF2 network in stromal cells would decrease the protective effects emanating from the microenvironment. To test this, shouldn't arms with "PD PD173074" alone, and with "nilotinib + PD173074" be included?

– why did you choose this model rather than a FLT3-ITD model? The jumping back and forth between CML and AML is distracting. Could the authors consider a focus on MOLM14 and FLT3-ITD driven AML in the main manuscript, with the CML work allocated to the supplement?

7) Please address the following clerical issues:

– abbreviations are defined several times (examples: TKI; conditioned media)

– conventions are not consistently followed (ml vs. mL; h vs. hrs vs. hours; spacing between value and unit [e.g. 10nM vs. 10 nM]; no spacing vs. spacing preceding a citation)

– Chemical formulas lack proper subscripting (possibly an inherent submission portal limitation)

– Please check supplier names (Genscript or GenScript? Is Jackson laboratories the correct name of this supplier?)

– References 14 and 30 are identical

– A few article titles in References section are title case (#4, 15, 16, for example)

– Reference 23 lacks page number, etc. The final manuscript is 2016, not 2015

– Reference 25 lacks journal name, etc.

– Figure 2 legend: NB40 or NP-40?

– Final paragraph of subsection “FGF2 is Contained in Stromal Cell ECVs and Exosomes”: "membrate"

– First paragraph of subsection “FGF2-*Fgfr1* Signaling Promotes Stromal Growth and Paracrine Protection of Leukemia”: "withg"

– "proteinase K" vs. "Proteinase K"

– "PD173074" vs. "PD-173074"

– Second paragraph of the Discussion: a callout to Figure 3E is made. Should this be 3D (there is no panel 3E)?

Additional data files and statistical comments:

I asked the authors to consider including additional data based on experiments with tyrosine kinase inhibitors beyond imatinib and AC220. If they choose to do this, inclusion of the data files is warranted. I do not see the need for any other additional data files.

Statistical analysis and description of statistical methods is adequate and appropriate.

Experimental rigor is evident in the work presented in the manuscript.

---

## [Author Response]

Reviewer #1:

[…] The authors conclude from the data in Figure 1C that ECVs produced by HS-5 cells are internalized by endocytosis. What evidence do the authors have the ECVs are actually entering cells by endocytosis and, if entry is by endocytosis, is it clathrin-mediated endocytosis, caveolae, micropinocytosis or phagocytosis?

The reviewer makes a good point, we do not present evidence that internalization is mediated by endocytosis. More detailed mechanistic studies of exosome transfer and uptake is in progress and beyond the scope of this manuscript. The text was modified as follows:

Analysis by confocal microscopy showed that ECVs are indeed internalized by leukemia cells, although the exact mechanism of internalization is still under investigation (Figure 1C and Supplemental movies).

The data in Figures2E/F on protection of exosomal FGF2 from proteinase K digestion are quite equivocal in that the partial protection observed is not entirely consistent with the authors' interpretations. Indeed, the protection described in Figure 2F is so marginal as to make this reviewer wonder how reproducible the data are and whether, or not, a modest increase in incubation time with proteinase K would lead to complete proteolysis of FGF2. Indeed, the authors' conclusions that, "….. FGF2 is contained within exosomes but also likely associated with proteins on the surface of exosomes." is rather ambivalent.

We repeated this experiment multiple times and it is technically challenging. Even the intact HS-5 cells in Figure 2E (mostly intracellular FGF2) have some degraded FGF2 after proteinase K treatment, likely from cell rupture. Isolated EVs and exosomes are even more fragile and we found that even integral membrane proteins tsg-101 and CD9 were partially degraded by proteinase K. The fact that we could detect intact FGF2 after proteinase K is strong evidence that some FGF2 (and perhaps most) is contained within EVs and exosomes. The text was modified to state this more explicitly.

We conclude that FGF2 is contained within EVs and exosomes, however we cannot exclude that FGF2 may also be present on the surface since partial FGF2 degradation was noted in intact HS-5 cells, EVs and purified exosomes (Figures 2E-F).

In Figure 3, the authors convincingly demonstrate that HS-5 CM contains more exosomes that HS-27 CM. They also note that the receptor for FGF2, FGFR1, is also present in exosomes. One obvious question is whether FGFR1 is tyrosine phosphorylated on sites that are stimulated by FGF2, which would suggest that a portion of FGF2 and FGFR1 are in complex with one another.

This is something we are quite interested in, however there are no reliable phospho-*FGFR1* antibodies that we have found – and not for lack of trying! To address the issue of FGF2 bound to *FGFR1* directly, we are generating FGF2 and *FGFR1* expression constructs with bi-fluorescent complementation tags. This system produces fluorescence when FGF2 and *FGFR1* are bound to each other, bringing the bi-fluorescent tags together to generate fluorescent signal. However, we do not have this data yet.

In Figure 5, the authors show that TKIs of FGFR1 signaling reduce exosome release from HS-5 stromal cells. This data are convincing, but it is also rather phenomenological and lacks a mechanism by which FGFR1 might regulate the release of exosomes by bone marrow stromal cells. Also, the fact that this occurs in one cell line does not lend confidence that this is a generalizable conclusion.

There are limited human marrow stromal cell lines available, but HS-5 and HS-27 are widely used as representative of distinct phenotypic subsets of marrow stroma and have clear differences in FGF2/*FGFR1* expression. We were able to replicate reduction in exosomes with FGFR inhibition in primary stroma from FGF2 +/+ mice but not FGF2 -/- mice (Figures 6 and 7), which we feel is the best genetic control. We attempted to treat primary human stroma with FGFR inhibitors but these cultures are quite heterogeneous making comparisons difficult, and we were unable to obtain sufficient conditioned media from primary cultures for reliable exosome quantification after FGFR inhibition.

This *FGFR1* downstream pathways that regulate exosome secretion is still under investigation in our lab. Some of our preliminary data is in Author response image 1 and suggests that *FGFR1* regulates exosome biogenesis through phospho-lipase C and protein kinase C activation, but experiments are ongoing to prove this.

Ongoing model and mechanistic work on PLC. We reasoned that *FGFR1* can either lead to increased exosome biosynthesis and/or release. One of the well-established FGFR signaling intermediaries is phospholipase C, a membrane-associated enzyme that cleaves phospholipids into diacyl glycerol (DAG) and inositol 1,4,5-triphosphoate (IP3). Secretory vesicle trafficking involves several steps that are controlled by DAG, including the fission of vesicles at the trans-Golgi network, the generation and maturation of multivesicular bodies, and the docking and fusion at the plasma membrane. Previous reports have shown that DAG kinase α (DGKα converts DAG to phosphatidic acid) expression decreases the production of FasL-containing exosomes by T lymphocytes, and that inhibition of DGKα enhances the production (Alonso et al. Cell Death Differ. 2011). DAG also activates protein kinase C (PKC), an enzyme which is recruited to the plasma membrane or to a number of intracellular compartments upon activation. PKC controls the endocytosis, trafficking and recycling of several receptor tyrosine kinases. This may be a relevant mechanism for the regulation of endocytosis and subsequent routing to multivesicular bodies. Although PKC has not previously been shown to phosphorylate *FGFR1, FGFR1* does possess C-terminal serine/threonine phosphorylation sites that are known to be important for regulation of endocytosis(Nadratowska-Wesolowska et al. Oncogene. 2014).

We therefore sought to test the roles of PLC and PKC in exosome release in HS-5 cells. A rise in intracellular calcium is necessary for regulated exocytosis of secretory granules and exosomes. We therefore used the cell-permeable calcium ionophore ionomycin as a positive control for the regulation of exosome release. PD173074, which we confirmed as a negative regulator of exosome production in previous figures, served as a negative control. Phorbol 12-myristate 13-acetate (PMA) is a mimic of DAG and activates PKC. Over a timecourse of 72 hours, both PMA and ionomycin led to increased exosome production/release. PD173074 attenuated exosome production. Interestingly, this was not rescued by combined treatment with ionomycin, suggesting that FGFR inhibits early exosome biogenesis rather than release.

In Figure 6, the use of Tet-regulated shRNA targeting shRNA is a good complement to the use of pharmacological TKIs of FGFR1 signaling. However, the effects of shFGFR1 on exosomal FGF2 should be assessed in Figure 6B – since the effects of shFGFR1 on tsg101 and CD9 are barely discernible given the poor quality of immunoblots presented in this figure. Furthermore, the authors need to better explain why their CRISPR/CAS9 efforts to silence the expression of either FGFR1 or FGF2 resulted in partial loss of FGFR1 or FGF2 expression. Moreover, whereas partial silencing of FGFR1 had almost no discernible effect on exosomal tsg101 or CD9, partial silencing of FGF2 had an appreciable effect on exosomal tsg101 and CD9. These data are somewhat discordant and are not adequately explained in the text.

The FGF2 protein in ECVs after shRNA silencing of *FGFR1* was assessed by immunoblot in this experiment, but unfortunately was uninterpretable. The amount of protein in ECVs is very low, and many of our immunoblots were at the limit of detection when performing small scale experiments (i.e. not using multiple 15 cm dishes of HS-5 cells to generate ECVs). The immunoblots for tsg-101 and CD9 in Figure 6 were darkened to allow better visualization of the bands.

The CRISPR/CAS9 experiments were complicated by the fact that complete silencing of either *FGFR1* or FGF2 in HS-5 cells essentially halted cell replication and we were never able to continue to culture cells with stable FGF2 and FGFR genetic deletion. We attempted this experiment numerous times, but could only analyze the cells for a short period after initial silencing. For this reason, the CRISPR/CAS9 cells only have partial silencing of FGF2 and *FGFR1* protein. We did notice that FGF2 silencing led to an increase in *FGFR1* protein on the cells and more rapid decrease in ECVs. The increase in *FGFR1* proteins is something we have observed with FGFR inhibitors previously (Traer et al. Blood 2014 and Traer et al. Cancer Research 2016) and based upon our previous data we suspect that removing FGF2 ligand leads to less activation and internalization of *FGFR1*. We suspect that silencing of *FGFR1* takes longer to affect ECV production since there is still abundant FGF2 ligand present to drive activity of the remaining *FGFR1*. Given the limitations of our genetic silencing tools, we could not investigate this more rigorously. Given the limitations of using CRISPR/CAS9 for FGF2 and *FGFR1* in HS-5 cells, this data was moved to the supplement, along with a more detailed explanation of the above issues.

In contrast to the CRISPR/CAS9 data, the primary stroma from FGF2 +/+ and -/- mice is a cleaner and more robust genetic control. Although the FGF2 -/- marrow stroma does not culture well in vitro, we could overcome this deficiency by simply using more mice. The immunoblots of ECVs from primary FGF2 +/+ and -/- stroma was moved to Figure 6 from the supplement.

We feel the presentation of the data this way is easier to understand, and the issues with the CRISPR/CAS9 system – although supportive overall of the model – does have technical limitations and is more appropriate for the Supplement.

The experiments presented in Figure 7 were nicely conceived and elegantly executed. This figure makes a case for an important role of stromal derived, exosome delivered FGF2 as a relevant mechanism of TKI resistance in leukemia.

We appreciate the positive feedback, thank you.

Reviewer #2:

The findings of the study are supported by strong, well-controlled experimental data. The paper is clearly written. My major concerns are the heavy reliance on HS-5 as proxy for the leukemic microenvironment and that the story is almost entirely presented from the perspective of the stroma. Several mechanistic questions pertaining to the leukemic cells are asked below, with the goal of helping the reader to better understand how the authors are viewing and rationalizing these points.1) A stated theme of the study is that the normal hematopoietic microenvironment is altered by leukemia. While this is undoubtedly correct, the choice of CM from a cell line derived from normal bone marrow would seem to limit applicability and relevance to the leukemic microenvironment.

Although HS-5 is derived from normal stroma, it has long been recognized has having distinct effects on hematopoiesis compared to HS-27. We previously evaluated FGF2 in stromal cells from patients with resistant CML and AML and found significantly increased FGF2 expression by immunohistochemistry during development of resistance (Traer et al. Blood 2014, Traer et al. Cancer Research 2016). The dramatically different expression of FGF2 and *FGFR1* in HS-5 compared to HS-27 make it a good model to study FGF2 secretion. Also, we were unable to generate sufficient numbers of ECVs from cultured primary stromal cells from leukemia patients to make this a feasible option using currently available methods for ECV quantification.

The data with primary murine stromal cells from FGF2 +/+ and -/- and the leukemia model in Figures 6 and 7 provides strong supportive data that marrow stromal FGF2 mediates resistance in the leukemia microenvironment specifically.

2) Why doesn't the presence of an FGFR inhibitor (PD173074) in experiments in which CM is provided to leukemic cell lines dampen the effectiveness of HS-5 CM? The authors make a convincing case that FGF2-laden exosomes are taken up by the leukemic cell lines and that this provides a protective effect against TKIs. They also show that FGFR inhibition (or [partial] FGFR1 deletion) in HS-5 stromal cells decreases exosome production/secretion and probably alters the content of exosomes. However, the story from the point of view of the leukemic cell lines is largely unexplored and mechanistically vague. Does FGFR1 deletion in the leukemic cell lines change the effectiveness of HS-5 CM? Does PD173074 completely shut off FGFR1 signaling in the leukemic cell lines? Perhaps I am missing something, but I find it surprising that inhibition of FGFR1 (in the leukemic cells) does not at least partially reverse the protective effects of HS-5 CM. This is largely based on the authors' reasonable assertion that the most important exosome cargo is FGF2.

Conditioned media from HS-5 cells contains a number of cytokines and other secreted factors in addition to FGF2-containing exosomes so the impact of adding an FGFR inhibitor such as PD173074 to CM is less pronounced in this setting (Figure 4A). If ECVs are isolated from HS-5 CM and then added to K562 or MOLM14 cells, then the inhibitory effect of PD173074 is more pronounced, although there is still some protection that is not blocked by FGFR inhibition. This is most likely due to protection from other components of exosomes/ECVs. To allow comparison with pure activation of FGFR, we did the same experiment with recombinant FGF2, in which case PD173074 completely blocks the protective effect. The data has been added to Figure 1—figure supplement 1.

The most impressive reduction in protection still comes from treating the HS-5 cells with PD173074 prior to collection of CM (Figure 4A). We found a similar reduction in protection with partial genetic silencing (Figure 6—figure supplement 2) and with genetic deletion of FGF2 from the leukemia microenvironment (Figure 7). This leads to reduced secretion of FGF2 in exosomes and likely has other effects on secretion of protective factors.

3) Several TKIs used in CML have target profiles that include FGFR, most notably the pan-FGFR TKI, ponatinib. Midostaurin, the currently approved frontline TKI for FLT3-mutated AML, inhibits FGFR1 and 2. Would patients treated with these TKIs derive additional benefit from collateral inhibition of FGFR1 in stromal cells (and possibly in leukemic cells)? I expected to see a comparison of imatinib and ponatinib in the study. I also thought this would be worth testing in the mouse model, as mentioned below.

We looked at ponatinib extensively in our previous CML publication (Traer et al. Blood 2014) and argue that additional FGFR inhibition contributes to the effectiveness of ponatinib in non-mutated, resistant CML patients. We have also found that midostaurin blocks many of the protective factors of stroma when MOLM14 cells are co-cultured with HS-5 cells (data not shown), similar to what we have previously published with quizartinib and PD173074 combination treatment (Traer et al. Cancer Research 2016). However, both ponatinib and midostaurin affect a number of additional pathways, especially at higher doses, so we chose to use specific inhibitors of FGFR in our experiments and the mouse model with deletion of FGF2 to get around the issue of off-target effects.

To further address this issue and minor point 1 below, we compared protection with recombinant FGF2, purified HS-5 ECVs, and CM with ECVs removed, using multiple ABL and FLT3 inhibitors. The data is now included in Figure 1—figure supplement 1. As predicted, higher doses of ponatinib and midostaurin are both able to overcome the protective effects of FGF2 and HS-5 ECVs at doses near the reported IC50 for FGFR (between 10-100 nM with ponatinib [Traer et al. Blood 2014] and 200nM midostaurin [Chen et al. PNAS 2014]).

4) Panels 4F,G show data for primary AML stromal cells (n = 42). The legend should specify how many AML patients are represented and how many data points are from each patient. Are some of these FLT3-ITD patients, and if so, can you indicate them as colored dots? This display item and the corresponding Materials and methods, Results and Discussion require more detail. This has potential to be an intriguing part of the story but the presentation is murky. Is it possible to include a comparison to HS-5 and HS-27 in these panels? Are the FGFR1 and FGF2 levels (and their correlations) substantially different from normal bone marrow? Should we expect them to be? Please consider strengthening this part of the manuscript. I could not glean much as presented.

There were nine patients with FLT3-ITD, and these samples are now indicated in Figure 4G, as well as more information about the stromal samples themselves. The primary stromal samples were collected from a variety of de novo and relapsed AML patients and therefore represents a heterogeneous mixture of AML patients at both diagnosis and relapse. There wasn’t a clear association of FGF2 and *FGFR1* overexpression with FLT3 ITD, genetic markers, or previous treatment; but there were not enough samples to be conclusive. We did not directly compare expression levels to HS-5 and HS-27 by QPCR.

In our previous publications, the amount of FGF2 in CML and AML marrow core biopsies by immunohistochemistry was not significantly increased compared to normal marrow prior to treatment, and FGF2 only became significantly increased after treatment (Traer et al. Blood 2014, Traer et al. Cancer Research 2016). Therefore we would not necessarily expect more FGF2 expression in stromal samples from patients at diagnosis. We are currently culturing stromal samples from AML patients treated with a FLT3 inhibitor as part of a clinical trial to evaluate change in FGF2 expression over time, but do not have this data yet.

To us, the important feature of the primary stromal expression data is the coordinate upregulation of FGF2 and *FGFR1* to drive autocrine activation in subsets of primary stromal samples. This would be predicted by our previous data, as well as published data on FGF2-*FGFR1* autocrine activation in murine marrow stromal cells during stress hematopoiesis (Itkin et al., 2012, Zhao et al., 2012). Finding evidence for this autocrine loop in primary cells indicates it can be activated under the correct conditions. We are very interested to discover what signals from treated leukemia cells stimulates FGF2 and *FGFR1* expression in marrow stroma, and this is currently under investigation.

The text was changed as follows:

“To evaluate FGF2 and *FGFR1* expression in primary leukemia stroma, bone marrow aspirates from a series of leukemia patients were cultured ex vivo and FGF2 and *FGFR1*-4 expression quantified by RT-PCR (Figure 4F). *FGFR1* and FGF2 transcripts were the most highly expressed in primary stroma, and there was a strong positive correlation between *FGFR1* and FGF2 expression (Figure 4G, r^2^=0.5683 and p<0.0001 on nonparametric correlation). This indicates that FGF2 and *FGFR1* expression are coordinately regulated in primary marrow stromal cells consistent with activation of an FGF2-*FGFR1* autocrine loop. There were 9 stromal cultures from AML patients with FLT3 ITD (indicated with red dots), but most of them were newly diagnosed, and based upon our previous data we would not expect increased expression of FGF2^16^. Similar to our observations in cell lines described above, we also detected *FGFR1* and FGF2 in ECVs derived from primary marrow stromal cultures (Figure 4—figure supplement 2A). However, primary marrow stromal cells grow slowly and produce smaller amounts of ECVs, so we were unable to evaluate the effect of FGFR inhibitors on cell morphology, growth, and ECV production with primary marrow stromal cells.”

Minor Comments:1) Figure 1: In panel A, what is the result if the same experiment is carried out with ponatinib (10-25 nM)? dasatinib (10-25 nM)? In panel B, with midostaurin (1 µM)? This comment is somewhat redundant with major point (3) above; here, I am requesting that the data are included, at least in the supplement.

Now included in Figure 1—figure supplement 1. Please see above for detail.

2) Why was 1 µM imatinib used? This concentration, at least at the 48 h timepoint, does not reduce proliferation (Figure 1A) to the extent that AC220 does in MOLM14 cells. Thus, it is difficult to determine the extent to which CM provides a protective effect. What happens if a higher concentration of imatinib is used, or if a more potent TKI is used at a concentration that matches the AC220 magnitude of reduced proliferation?

We used 1 µM imatinib primarily for comparison with our previous experiments. Now included in Figure 1—figure supplement 1 are a variety of ABL inhibitors and concentrations. Even with higher concentrations and more potent inhibitors K562 cells undergo apoptosis more slowly than MOLM14 cells.

3) Regarding the FGFR inhibitors:– Why two? BGJ-398 is described as 'selective' but I did not find the rationale for needing it in the study (which began with PD173074).– What is the direct evidence/readout to confirm that these inhibitors are working (and to measure the extent of inhibition)?

Previous reviewers requested experiments with more selective FGFR inhibitors and BGJ-398 is one of the newer and more selective inhibitors.

4) Perhaps it's not a reasonable comparison, but it is surprising that shRNA-based elimination of FGFR1 does not reduce FGF2 levels in HS-5 cells (Figure 6A), but FGFR inhibition does (Figure 5C). Also, the HS-5 CRISPR FGFR1.1 and 1.2 lines show reduced FGF2. The HS-5 CRISPR FGF2.1 and 2.2 lines show enhanced FGFR1 but reduced FGFR1 in ECVs. Can the authors comment on / explain these points?

We suspect most of the variability with FGF2 levels after shRNA or CRISPR/CAS of *FGFR1* or FGFR inhibition are assay specific and related to when the cells were evaluated by immunoblot. Reviewer #1 also commented on this, and a more complete discussion is outlined above.

5) In Figure 6B, I would like to be able to evaluate whether reduced FGFR1 brings about a reduction in FGF2, but this data is not provided. In general, both FGF2 and FGFR1 should be shown in immunoblot stacks. Sometimes, it is one or the other.

Reviewer #1 also raised this issue and the absence of FGF2 staining in that figure was a technical issue (see above). Since we often had limited supplies of ECVs, we had to run the entire sample to obtain enough signal for detection, which did not allow for repeat immunoblots.

6) Figure 7:– why is nilotinib used in 7A? All other data are with imatinib.– the main point of the paper seems to be that interrupting the FGFR1-FGF2 network in stromal cells would decrease the protective effects emanating from the microenvironment. To test this, shouldn't arms with "PD PD173074" alone, and with "nilotinib + PD173074" be included?– why did you choose this model rather than a FLT3-ITD model? The jumping back and forth between CML and AML is distracting. Could the authors consider a focus on MOLM14 and FLT3-ITD driven AML in the main manuscript, with the CML work allocated to the supplement?

We used the more potent inhibitor nilotinib to allow once daily gavage. This still allows for disease suppression and reduces trauma to the mice (and lab technicians). Imatinib is usually given twice daily by gavage.

The combination of an FGFR inhibitor and ABL inhibitor would be predicted to also reduce resistance, but the FGF2 -/- is a cleaner genetic model. We are currently trying to find pharma support to test relevant combinations with an FGFR inhibitor in clinical development.

The BCR-ABL model is a well-characterized model in the lab and does not require breeding. The FLT3-ITD and TET2 mouse model that we have access to would require extensive back-breeding with the FGF2 +/+ and -/- mice, adding significantly more time and expense.

We feel the focus of this manuscript on marrow stroma provides a unifying theme for our previous discoveries that FGF2 promotes resistance in both CML and AML. Our data also suggests that targeting the stroma directly with FGFR inhibitors is a novel therapeutic target in multiple leukemias.

7) Please address the following clerical issues:– abbreviations are defined several times (examples: TKI; conditioned media)– conventions are not consistently followed (ml vs. mL; h vs. hrs vs. hours; spacing between value and unit [e.g. 10nM vs. 10 nM]; no spacing vs. spacing preceding a citation)– Chemical formulas lack proper subscripting (possibly an inherent submission portal limitation)– Please check supplier names (Genscript or GenScript? Is Jackson laboratories the correct name of this supplier?)– References 14 and 30 are identical– A few article titles in References section are title case (#4, 15, 16, for example)– Reference 23 lacks page number, etc. The final manuscript is 2016, not 2015– Reference 25 lacks journal name, etc.– Figure 2 legend: NB40 or NP-40?– Final paragraph of subsection “FGF2 is Contained in Stromal Cell ECVs and Exosomes”: "membrate"– First paragraph of subsection “FGF2-FGFR1 Signaling Promotes Stromal Growth and Paracrine Protection of Leukemia”: "withg"– "proteinase K" vs. "Proteinase K"– "PD173074" vs. "PD-173074"– Second paragraph of the Discussion: a callout to Figure 3E is made. Should this be 3D (there is no panel 3E)?

Our apologies for the errors and inconsistencies. This has been corrected, although I left a definition of conditioned media in both the text and Figure 1 legend since I like to look at figures and legends first.

Additional data files and statistical comments:I asked the authors to consider including additional data based on experiments with tyrosine kinase inhibitors beyond imatinib and AC220. If they choose to do this, inclusion of the data files is warranted. I do not see the need for any other additional data files.

The Prism data files were uploaded.